# CATASTROPHIC NEGATIVE TRANSFER: AN OVERLOOKED PROBLEM IN CONTINUAL REINFORCEMENT LEARNING

## ABSTRACT

Continual Reinforcement Learning (CRL) recently witnessed significant advancements, but negative transfer, a phenomenon in which policy training for new task fails when trained after a specific previous task, has been largely overlooked. In this paper, we shed light on the prevalence and catastrophic nature of the negative transfer in CRL through systematic experiments on the Meta-World RL environments. Our findings highlight that this phenomenon possesses a unique characteristic distinct from the mere reduction in plasticity or capacity observed in conventional RL algorithms. Then, we introduce a simple yet effective baseline called Reset & Distill (R&D) to address the issue of negative transfer in CRL. R&D combines a strategy of resetting the agent's online actor and critic networks to learn a new task and an offline learning step for distilling the knowledge from the online actor and previous expert's action probabilities. As a result, our method can successfully mitigate both catastrophic negative transfer and forgetting in CRL. We carried out extensive experiments on long sequence of Meta-World tasks and show that our method consistently outperforms recent baselines, achieving significantly higher success rates across a range of tasks. Our findings highlight the importance of considering negative transfer in CRL and emphasize the need for robust strategies like R&D to mitigate its detrimental effects.

## 1 INTRODUCTION

Following the impressive recent success of reinforcement learning (RL) (Mnih et al. (2013); Silver et al. (2016); Mnih et al. (2015); Andrychowicz et al. (2020)) in various applications, a plethora of research has been done in improving the learning efficiency of RL algorithms. One important avenue of the extension is the Continual Reinforcement Learning (CRL), in which an agent aims to continuously learn and improve its decision-making policy over sequentially arriving tasks without forgetting previously learned tasks. The motivation for such extension is clear since it is not practical to either re-train an agent to learn multiple tasks seen so far or train a dedicated agent for each task whenever a new task to learn arrives. The need for CRL is particularly pressing when the sequentially arriving tasks to learn are similar to each other as in robot action learning (Kober et al. (2013)).

In general, one of the main challenges of continual learning (CL) is to effectively transfer the learned knowledge to a new task (*i.e.*, improve plasticity) while avoiding catastrophic forgetting of previously learned knowledge (*i.e.*, improve stability). So far, most of the CRL methods (Mendez et al. (2020; 2022); Rolnick et al. (2019); Wolczyk et al. (2022)) also focus on addressing such a challenge, largely inspired by the methods developed in the supervised learning counterparts such as regularization-based methods (Kirkpatrick et al. (2017); Zenke et al. (2017); Ahn et al. (2019); Jung et al. (2020); Schwarz et al. (2018)), memory-based methods (Chaudhry et al. (2019b;a); Lopez-Paz & Ranzato (2017)), and parameter isolation methods (Mallya & Lazebnik (2018); Mallya et al. (2018); Hung et al. (2019); Yoon et al. (2018)).

Due to the aforementioned trade-off, it is generally understood that the plasticity degradation in continual learning occurs when the learner tries to increase the stability. However, several recent works pointed out that regardless of the degree of the stability, the plasticity of the learner can still decrease drastically even when learning a *single* task due to the nature of the learning procedure

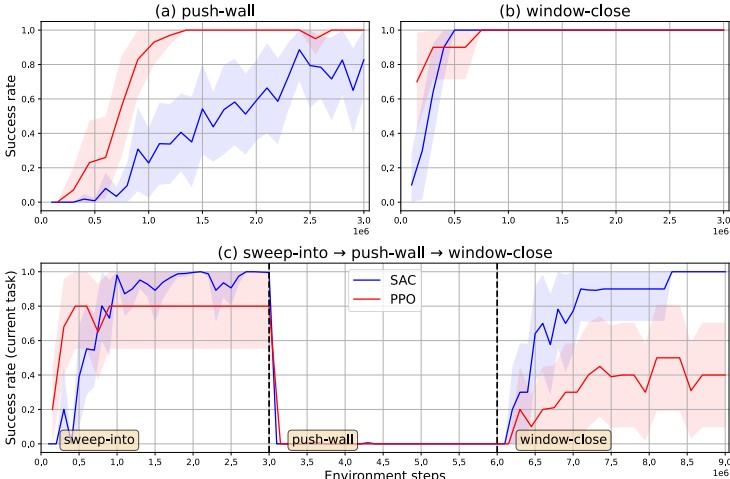

Figure 1: Success rates of SAC (blue) and PPO (red) while (a) learning `push-wall` task from scratch, (b) learning `window-close` task from scratch, and (c) continuously learning `sweep-into`, `push-wall`, and `window-close` tasks.

in RL. For example, Nikishin et al. (2022) demonstrated that RL methods which tend to highly over-fit to the initial data in the replay buffer can suffer from *primacy bias* that leads to the plasticity degradation for the incoming samples. Similarly, Lyle et al. (2022); Kumar et al. (2021) showed that since the target for learning a value function becomes non-stationary as learning progresses, the rank of the feature embeddings in the value network eventually shrinks, which results in the *capacity loss* of the value function and hinders the function from learning new tasks. In addition, several works also pointed out a similar phenomenon in the context of CRL. Namely, Dohare et al. (2021) considered that the agent learned with stochastic gradient descent (SGD), which does not take the stability into account, also suffers from the plasticity degradation during CRL. Moreover, Abbas et al. (2023) showed via extensive experiments that when a learner continuously learns a sequence of RL tasks multiple times via fine-tuning, the performance on each task would degrade as the learning progresses. The authors dubbed this phenomenon as *plasticity loss* of a learner, which is essentially very similar to the capacity loss mentioned above; namely, the RL methods suffer from plasticity degradation even when the stability is not considered.

In this paper, we consider a similar, but significantly different, *catastrophic negative transfer* problem in CRL — in which a learner suffers from a severe plasticity degradation *depending* on the learned task sequence. Specifically, through a series of experiments, we demonstrate that the level of such degradation while learning a newly arrived task (with fully plastic fine-tuning) is seriously affected by the nature or the existence of the preceding task. Such dependency on the previous tasks turns out to be quite substantial to the extent that a learner may *completely fail* in learning a new task (hence, "catastrophic" negative transfer). This failure is particularly noteworthy considering the same task might have been easily trainable had the learner been exposed to a different preceding task or started anew from scratch. We stress that all the related works regarding the capacity/plasticity loss of an RL agent have overlooked this significant interplay among different tasks in CRL.

To showcase the catastrophic negative transfer problem more concretely, we carry out a simple experiment on a popular Meta-World (Yu et al. (2020)) environment. Figure 1(a) and (b) show the success rates for learning the `push-wall` and `window-close` tasks with SAC (Haarnoja et al., 2018) and PPO (Schulman et al., 2017) algorithms for 3 million (M) steps, respectively. Note that both algorithms quickly achieve success rate close to 1, showing both `push-wall` and `window-close` are quite easy to learn from scratch. Now, Figure 1(c) shows the results for continuously learning `sweep-into`, `push-wall` and `window-close` tasks with simple *fine-tuning* for 9M steps (3M steps for each task). In the figure, we can clearly observe that both algorithms *completely fail* to learn the second task, `push-wall`, although it can be easily learned from scratch as shown in Figure 1(a). One may hypothesize that such a failure could be simply due to the capacity/plasticity loss of an RL agent as argued in Lyle et al. (2022); Kumar et al. (2021) Abbas et al. (2023). If it were the case, the learner should also not be able to learn the third task, `window-close`, since there would be no remaining network capacity to learn a new task. However, as can be seen in Figure 1(c), both

algorithms can effectively learn `window-close`, which suggests that failing to learn `push-wall` may not be due to the simple plasticity loss, but due to the catastrophic negative transfer from the `sweep-into` task.

To make our observation and claim more formal, in section 3, we extensively analyze the two-task CRL scenario among various tasks in Meta-World. We show that the catastrophic negative transfer indeed occurs quite often, regardless of the learning algorithms. Subsequently, in section 4, we propose a simple yet effective method, dubbed as **R**eset and **D**istill (**R&D**), which can prevent both catastrophic negative transfer and catastrophic forgetting, thus, achieving a superb CRL performance. Our thorough experimental results in section 5 show that R&D can outperform other recent CRL baselines by significant margin on a longer sequence of tasks.

## 2 PRELIMINARIES

### 2.1 CONTINUAL REINFORCEMENT LEARNING (CRL)

In CRL, an agent needs to sequentially learn multiple tasks without forgetting the past tasks. We denote the task sequence by a task descriptor $\tau \in \{1, ..., T\}$, in which $T$ is the total number of tasks. At each task $\tau$, the agent interacts with the environment according to a Markov Decision Process (MDP) $(\mathcal{S}_\tau, \mathcal{A}_\tau, p_\tau, r_\tau)$, where $\mathcal{S}_\tau$ and $\mathcal{A}_\tau$ are the set of all possible states and actions for task $\tau$. Given $s_{t+1}, s_t \in \mathcal{S}_\tau$ and $a_t \in \mathcal{A}_\tau$, $p_\tau(s_{t+1}|s_t, a_t)$ is the probability of transitioning to $s_{t+1}$ given a state $s_t$ and action $a_t$. $r_\tau(s_t, a_t)$ is the reward function that produces a scalar value for each transition $(s_t, a_t)$. The objective of an RL agent is to obtain a policy $\pi(a_t|s_t)$ that can maximize the sum of expected future rewards for each task $\tau$.

In this paper, we focus mainly on the actor-critic method that combines both value-based and policy-based methods. The method includes two networks: an **actor** that learns a policy and a **critic** that learns the value function; the critic evaluates the policy by estimating the value of each state-action pair, while the actor improves the policy by maximizing the expected reward. Given task $\tau$ and $s_t \in \mathcal{S}_\tau, a_t \in \mathcal{A}_\tau$, we denote the actor parameterized by $\boldsymbol{\theta}_\tau$ as $\pi(a_t|s_t; \boldsymbol{\theta}_\tau)$, and the critic parameterized by $\boldsymbol{\phi}_\tau$ as $Q(a_t, s_t; \boldsymbol{\phi}_\tau)$. For the algorithms that only use the state information in the critic (Schulman et al. (2017)), we denote the critic network as $V(s_t; \boldsymbol{\phi}_\tau)$. After learning all $T$ tasks, the main objective of the actor-critic based CRL is to obtain an actor network $\pi(\cdot|\cdot; \boldsymbol{\theta}_T)$ which can do well on all tasks seen so far without catastrophic forgetting. There are several well-established actor-critic methods, *e.g.*, SAC (Haarnoja et al. (2018)) and PPO (Schulman et al. (2017)), and for further details, we refer to the original papers.

### 2.2 BEHAVIORAL CLONING

Behavioral Cloning (BC) (Pomerleau (1988)) is a technique used in imitation learning that let the learning policy mimic the behavior of an expert policy. In CRL, (Wolczyk et al. (2022)) first adopted BC to prevent catastrophic forgetting and showed that their method outperforms the state-of-the-art baselines. More concretely, their method constructs an *expert buffer* $\mathcal{M}^k$, which contains the pairs of state and action distribution learned by the policy, $(s_t, \pi(\cdot|s_t; \boldsymbol{\theta}_k))$, for each previous task $k$. Then, an agent learning a new task $\tau$ adds the following Kullback-Leibler (KL) divergence terms as a regularizer to the original RL loss:

$$\ell_{\text{BC}}(\boldsymbol{\theta}_\tau) = \sum_{(s_t, \pi(\cdot|s_t; \boldsymbol{\theta}_k)) \in \mathcal{M}_\tau} \text{KL}(\pi(\cdot|s_t; \boldsymbol{\theta}_\tau)||\pi(\cdot|s_t; \boldsymbol{\theta}_k)), \tag{1}$$

in which $\mathcal{M}_\tau = \bigcup_{k=1}^{\tau-1} \mathcal{M}^k$ is the union of all expert buffers up to task $\tau - 1$. Thus, the BC term regularizes the new actor network to have similar action distributions as those of old actor networks. Moreover, note that since $\pi(\cdot|s_t; \boldsymbol{\theta}_k)$'s are stored in $\mathcal{M}^k$ for each task $k$, we need not store the actual parameters $\boldsymbol{\theta}_k$ of the past actors.

## 3 CATASTROPHIC NEGATIVE TRANSFERS IN CRL

To identify the negative transfer problem in CRL more generally, we carried out extensive experiments using Meta-World (Yu et al. (2020)) environment, which consists of 50 robotic manipulation tasks.

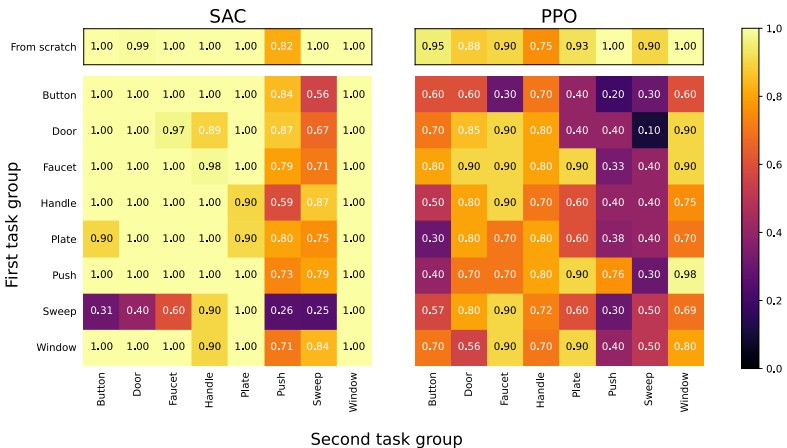

Figure 2: The results on training SAC (left) and PPO (right) on all task groups. The top rows represent the average success rates on all 24 tasks averaged over its corresponding group and the random seeds. The bottom matrix shows the average success rate of the second task when the algorithms are fine-tuned after learning the first task. All experiments were averaged over 10 random seeds.

First, we carefully selected 24 tasks that can be learned from scratch within 3M steps. We then categorized them into 8 groups by grouping the tasks that share the same first word in their names, assuming they are similar to each other. The 8 task groups were {`Button`, `Door`, `Faucet`, `Handle`, `Plate`, `Push`, `Sweep`, `Window`}, and for more details on the specific tasks in each group, please refer to the Supplementary Material.

From those task groups, we carried out extensive two-task CRL experiments as in Figure 1 on *all* (*i.e.,* 64) pairs of task groups. Specifically, for each pair of distinct task groups, we randomly sampled two specific tasks from the corresponding groups, and for each pair of identical task group, we randomly sampled two *different* tasks from the group. On those 64 pairs of two tasks, we again employed SAC (Haarnoja et al. (2018)) and PPO (Schulman et al. (2017)) and checked whether the fine-tuning to the second task indeed suffers from the catastrophic negative transfer. Furthermore, to compare the results of the tasks in the second tasks group with single task performance, we also trained both algorithms on all 24 tasks from scratch. We also conduct experiments that present the task-wise negative transfer results consisting of 13 tasks, and the results are in the Supplementary Materials. All the experiments were done with 10 different random seeds and the success rates for the second task after learning were averaged over the number of episodes and random seeds.

Figure 2 shows the average success rate of each group when training from scratch (top row) and of the second task group learned by fine-tuning after learning the first task (bottom matrix) for both SAC and PPO. From the top rows in the figure, we observe that both SAC and PPO can learn most of the tasks from scratch well, since most of the average success rates are close to 1. However, in the bottom matrices, we observe that a significant portion of the second tasks are hard to learn when the first task was learned before fine-tuning on the second task (with PPO being more severe). For example, for SAC, when the first task is from the `Sweep` group, tasks from `Button`, `Door`, `Faucet`, `Push`, and `Sweep` groups are hard to learn when they come as the second task , while most of them are easy to learn when the first task is from other groups. For PPO, when the first task is from `Button` or `Sweep` group, most of the second tasks were hard to learned. Moreover, for both algorithms, we observe that the tasks in `Push` and `Sweep` group were hard to learn as a second task regardless of the type of the first task, implying that the tasks in both groups are always prone to the catastrophic negative transfer from the first task. Furthermore, note the diagonal of the matrix contains the results when the first task was also selected from the same group[1]. For that case, we observe SAC can learn second tasks well except for the `Sweep` group, while PPO could learn the second tasks well only for 2 task groups (*e.g.*, `Faucet`, `Handle`). Therefore, even between the tasks belonging to the same task group can cause the negative transfer.

---

[1]One should not confuse these results with those for transferring to the same *task*.

---

**Algorithm 1** R&D algorithm

---

**Require:** $\theta_{\text{Online}}$ and $\phi_{\text{Online}}$: Randomly initialized network for learning single task in online way
**Require:** $E$: Number of epochs, $T$: Total number of tasks
   **for** $\tau = 1, \cdots, T$ **do**
      Randomly initialize $\theta_{\text{Online}}$ and $\phi_{\text{Online}}$ /* Reset the online actor */
      Learn task $\tau$ using $\theta_{\text{Online}}$ and $\phi_{\text{Online}}$, and produce the replay buffer $\mathcal{D}_\tau$
      **for** $e = 1, \cdots, E$ **do**
         **for** $\mathcal{B}_{\mathcal{D}_\tau} \sim \mathcal{D}_\tau$ **do**
            $\mathcal{B}_{\mathcal{M}_\tau} \sim \mathcal{M}_\tau$
            Compute $\ell_{\text{R\&D}}(\theta_\tau)$ using $\mathcal{B}_{\mathcal{D}_\tau}$ and $\mathcal{B}_{\mathcal{M}_\tau}$ /* Distillation from $\theta_{\text{Online}}$ */
            Optimize $\theta_\tau$ using SGD
         **end for**
      **end for**
      Store small subset of $\mathcal{D}_\tau$ into $\mathcal{M}_\tau$
   **end for**

---

In summary, we clearly observe that the learnability of the second task is highly dependent on the type or the existence of the first task, which cannot be simply explained by the plasticity/capacity loss hypotheses of prior work. To that end, developing a novel method for addressing this catastrophic negative transfer problem is necessary.

## 4 R&D: A SIMPLE BASELINE FOR ADDRESSING NEGATIVE TRANSFER

In this section, we propose a simple yet strong baseline that can resolve not only the catastrophic negative transfer in CRL, while also addressing the catastrophic forgetting problem. In Figure 2, we observed that the single tasks, which are learned from randomly initialized actor and critic networks, were all easy to learn, but when the networks are initialized with those learned from the first task (for fine-tuning), many of those tasks were not learnable. This result suggests that the reward signals (from the second task), unlike the supervised labels, are not strong enough to correctly modify the network parameters to successfully adapt to the given task.

From this reasoning, we propose to use two kinds of actor networks for CRL. One is an online actor network that mainly contributes to learning new task in an online manner by interacting with the environment. The other is an offline actor that clones the behaviour of the online actor in an offline manner without any interaction with environment. More concretely, let us denote the parameters of online actor and critic network as $\theta_{\text{Online}}$ and $\phi_{\text{Online}}$, respectively. First, the online actor and critic independently learns a task $\tau$ *from scratch*, and the learning algorithm produces the replay buffer $\mathcal{D}_\tau$ [2]. Then, using the state information in $\mathcal{D}_\tau$, the online actor distills the knowledge of the new task to the offline actor. At the distillation phase, to prevent the catastrophic forgetting, the offline actor also uses the expert buffer $\mathcal{M}_\tau$. Hence, the loss function for the offline actor $\theta_\tau$ becomes

$$\ell_{\textbf{R\&D}}(\theta_\tau) = \underbrace{\sum_{s_t \in \mathcal{B}_{\mathcal{D}_\tau}} \text{KL}(\pi(\cdot|s_t; \theta_{\text{Online}}) || \pi(\cdot|s_t; \theta_\tau))}_{(a)} + \underbrace{\sum_{(s_t, \pi(\cdot|s_t; \theta_k)) \in \mathcal{B}_{\mathcal{M}_\tau}} \text{KL}(\pi(\cdot|s_t; \theta_\tau) || \pi(\cdot|s_t; \theta_k))}_{(b)},$$

in which $\mathcal{B}_{\mathcal{M}_\tau}$ and $\mathcal{B}_{\mathcal{D}_\tau}$ are mini-batch sampled from $\mathcal{M}_\tau$ and $\mathcal{D}_\tau$, respectively. Note term (a) is used to distill the knowledge from the online actor (hence, promotes plasticity), and term (b) is exactly the same as that for BC (1) (hence, promotes stability). Thus, the complexity of the offline actor distillation is essentially on par with that of supervised learning, introducing just a minor additional computational workload compared to the online actor and critic's RL task learning process.

After the distillation phase, the $\theta_{\text{Online}}$ and $\phi_{\text{Online}}$ are reset to learn the next task from scratch so that the catastrophic negative transfer can be fundamentally prevented. Consequently, we dub our algorithm as **R**eset and **D**istill (**R& D**), and the overall summary of our algorithm is in Algorithm 1.

---

[2]For algorithms that do not have the replay buffer, *e.g.*PPO (Schulman et al., 2017), we instead sample large amount of episodes after training and use them.

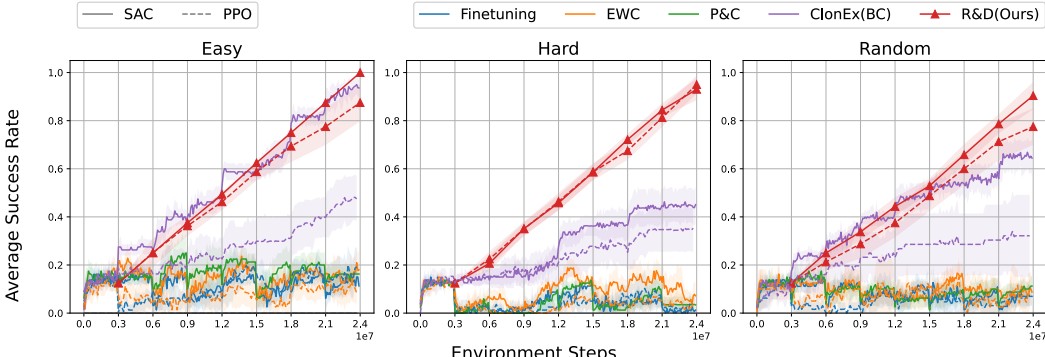

Figure 3: The average success rates of different methods were assessed across three types of sequences: 'Easy', 'Hard', and 'Random'. Solid and dashed lines denote results obtained using SAC and PPO, respectively. Each graph is color-coded to indicate the specific CL method applied. Note that because of the severe negative transfer when training P&C with PPO, the loss diverges to infinity, so that we are unable to train P&C with PPO. Therefore, we only report the results of P&C with SAC.

*Remark 1:* The rationale for our method is that since the reward signals are not strong enough to always adapt the pre-trained actor and critic networks to the current task, we first learn the current task from scratch then treat the action distribution of the learned online actor as the ground-truth supervised label for the offline actor.

*Remark 2:* One may think instead of completely initializing the networks, only initializing either actor or critic for learning the new task and combining with BC would do the job. However, as we show in the ablation study, such partial initialization would not fully address the negative transfer.

## 5 EXPERIMENTAL EVALUATION

First, we evaluated R&D on long task sequences which consist of multiple environments from Meta-World, and compare the results with several state-of-the-art CL baselines. Then, to show whether the methods developed for resolving the capacity loss (Lyle et al. (2022); Kumar et al. (2021)) or the plasticity loss (Abbas et al. (2023)) can also resolve the catastrophic negative transfer, we carried out same experiments as those conducted in Section 3. Specifically, Lyle et al. (2022) proposed the feature regularization technique, dubbed as InFeR, to secure the feature rank of the value function and Abbas et al. (2023) proposed that using Concatenated ReLU (CReLU) can prevent the plasticity loss. Utilizing those methods, we fine-tuned an agent on the two consecutive tasks and also evaluated those methods on long task sequences. Furthermore, to show the effectiveness of R&D on resolving both the catastrophic negative transfer and forgetting, we carried out several analyses on measuring the negative transfer and forgetting in long sequence experiments. Lastly, we investigated the effect of the size of replay buffer and expert buffer in our method. In all the experiments, we again used SAC and PPO. For more details on the experiment settings, please refer to the Supplementary Materials.

### 5.1 EVALUATION ON LONGER SEQUENCE

For the experiment, we used a total of 3 task sequences. Firstly, we identified task pairs that exhibit negative transfer when fine-tuning two tasks consecutively. With this information, it is possible to compare the potential difficulties between the task sequences we want to learn. For example, consider different task sequences like A→B→C→D and E→F→G→H where each alphabet represents one task. If we observed negative transfer occurring in consecutive task pairs (A, B), (C, D) and (F, G) within the sequences, the first sequence contains two pairs likely to exhibit negative transfer, while the second has only one such pair. Therefore we can expect the first sequence to be more challenging than the second.

We utilized this method to create two task sequences, each with a length of 8: 'Hard' and 'Easy'. The 'Hard' sequence comprises 6 task pairs where negative transfer occurs in the 2-task setting, while the 'Easy' sequence is generated by connecting only those task pairs where negative transfer does not occur. To further validate the results in an arbitrary sequence, we randomly chose 8 out of the

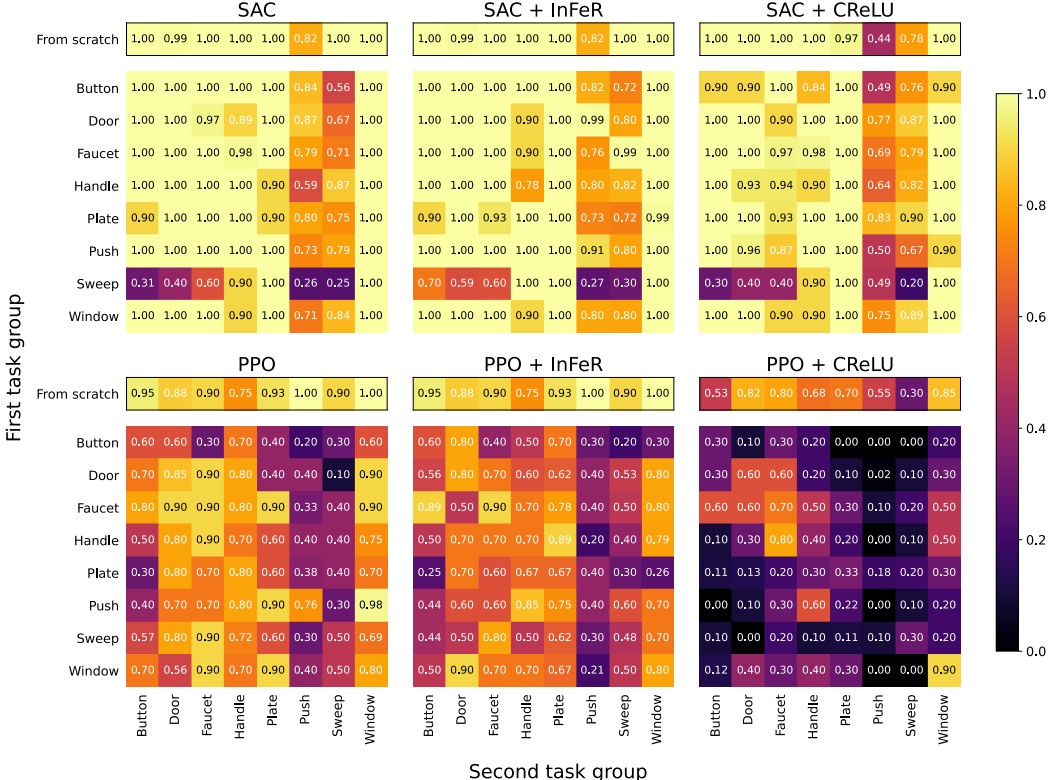

Figure 4: Two-task CRL experiments on various methods. Note that for the methods with CReLU, the results of 'From scratch' are obtained by training vanilla RL methods with CReLU.

24 tasks employed in the preceding section and conducted training by shuffling them based on each random seed. Henceforth, we will refer these arbitrary sequences as the 'Random' sequence.

The sequences constructed using as above are as follows. Task name marked in bold-italic indicates that negative transfer may occur when it is learned continuously followed by the previous task.

- Easy: `faucet-open` → `door-close` → `button-press-topdown-wall` → `handle-pull` → `window-close` → `plate-slide-back-side` → `handle-press` → `door-lock`

- Hard: `faucet-open` → ***push*** → ***sweep*** → ***button-press-topdown*** → `window-open` → ***sweep-into*** → ***button-press-wall*** → ***push-wall***

- Random: `door-unlock, faucet-open, handle-press-side, handle-pull-side, plate-slide-back-side, plate-slide-side, shelf-place, window-close`

In this experiment, we compared our method to three CL baselines: Elastic Weight Consolidation (EWC) (Kirkpatrick et al. (2017)), Progress & Compress (P&C) (Schwarz et al. (2018)), and ClonEx (Wolczyk et al. (2022)), along with naïve fine-tuning. Please note that ClonEx leverages the best-reward exploration technique originally designed only for SAC, leading us to choose Behavioral Cloning (BC) as the method for PPO implementation. Among these methods, P&C is similar to our method in terms of distilling the knowledge obtained from the fine-tuned model into the continual learner. However, there are two key differences between P&C and our method. Firstly, rather than utilizing an expert buffer, P&C employs EWC to mitigate catastrophic forgetting in the knowledge base model. Secondly, P&C employs only one active column throughout the entire task sequence without any re-initialization. Hence, P&C can be vulnerable to the potential adverse effects of negative transfer when it undertakes the sequential learning of eight distinct tasks.

Figure 3 shows the results of various methods on 3 different sequences. Since the offline actor of R&D learns new tasks in offline way, we instead put markers on the results of R&D and connected them with lines to notice the difference between the baselines. All results are averaged over 10 random seeds. In this figure, we can observe that R&D outperforms all baselines across all sequences

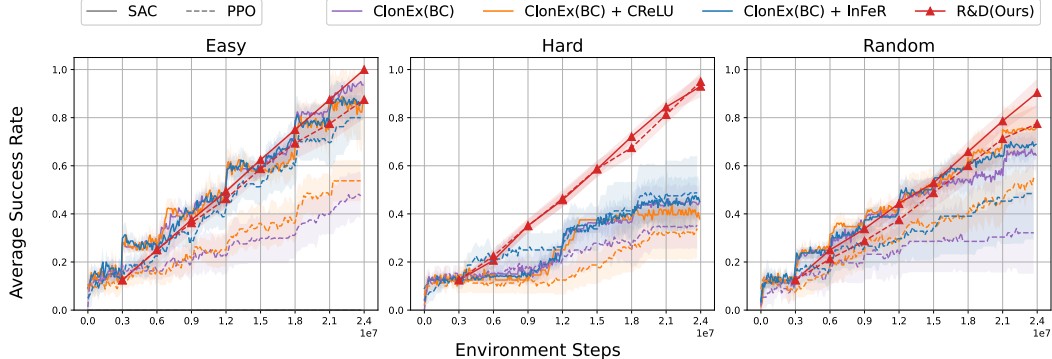

Figure 5: The results of long sequence experiment on ClonEx with CReLU or InFeR. To ease the comparison between schemes, we again show the results on R&D and vanilla ClonEx.

by a significant margin. Additionally, the average success rate of R&D is nearly 1, indicating its success in overcoming both catastrophic forgetting and negative transfer.

## 5.2 CAN CRELU OR INFER RESOLVE THE CATASTROPHIC NEGATIVE TRANSFER?

Next, we fine-tuned SAC and PPO equipped with CReLU (Abbas et al. (2023)) or InFeR (Lyle et al. (2022)) on two consecutive tasks to investigate their potential to mitigate negative transfer effects, as detailed in section 3. Figure 4 provides the results. It is worth noting that since applying CReLU changes the architectural properties of the networks, we should start from the first task again, potentially leading to a shift in results 'From scratch'. The findings reveal that fine-tuning with CReLU and InFeR yields similar success rates when compared to the results presented in Figure 2. Furthermore, learning tasks from scratch using CReLU achieves much lower success rates than the results from naive SAC and PPO. Therefore, we suggest that solutions such as CReLU or InFeR, aimed at addressing plasticity or capacity loss, may not be suitable for preventing negative transfer.

In the following experiment, we will show whether the results are consistent in the long task sequence experiments. To check that either CReLU or InFeR can be compatible to the CL baseline, we combined the ClonEx which is a representative baseline among others with CReLU or InFeR. Figure 5 shows the results. As we expected, except for ClonEx with PPO, all the other schemes cannot improve the performance of ClonEx since both CReLU and InFeR are not able to prevent the catastrophic negative transfer. Therefore, the success rates of the hybrid methods are still much lower than R&D with significant margin.

## 5.3 ANALYSES ON NEGATIVE TRANSFER AND FORGETTING

To quantitatively analyze how negative transfer and forgetting actually occurs in our experiments, we measured the forgetting and negative transfer of 3 methods: R&D, Fine-tuning, and ClonEx(BC). Furthermore, to quantify how much CReLU or InFeR resolves the catastrophic negative transfer, we also measure the forgetting and negative transfer of ClonEx with CReLU or InFeR. Let us denote the success rate of the task $j$ when the actor immediately finished learning task $i$ as $R_{i,j}$, and the success rate after training task $i$ from scratch as $R_i^{\text{Single}}$. Then the negative transfer after learning task $\tau$, denoted as $NT_\tau$, and the forgetting of task $i$ after learning task $\tau$, denoted as $F_{\tau,i}$, are defined as follows, respectively:

$$NT_\tau = R_\tau^{\text{Single}} - R_\tau \quad \text{and} \quad F_{\tau,i} = \max_{l \in \{1,...,i-1\}} R_{l,i} - R_{\tau,i}. \tag{2}$$

After learning all $T$ tasks, for the negative transfer, we report the average of $NT_\tau$ for all task $\tau \in \{1, ..., T\}$, and for the forgetting, we report the average of $F_{T,i}$ for task $i \in \{1, ..., T-1\}$. Table 2 shows the results on the negative transfer and forgetting measured by (2). Note that the lower values of both negative transfer and forgetting are better in our setting. In this table, all CRL baselines, except for R&D, display vulnerability to negative transfer. Across all methods, negative transfer tends to be more prominent in the 'Hard' sequence compared to the 'Easy' sequence, whereas it appears

Table 1: The results on negative transfer and forgetting with various schemes.

| Measure | Negative transfer (↓) | | | Forgetting (↓) | | |
|---|---|---|---|---|---|---|
| Sequence | Easy | Hard | Random | Easy | Hard | Random |
| | SAC / PPO | | | | | |
| Fine-tuning | 0.096 / 0.379 | 0.500 / 0.624 | 0.193 / 0.425 | 0.900 / 0.361 | 0.504 / 0.331 | 0.777 / 0.336 |
| EWC | 0.071 / 0.536 | 0.457 / 0.676 | 0.260 / 0.375 | 0.852 / 0.319 | 0.512 / 0.281 | 0.671 / 0.430 |
| P&C | 0.071 / - | 0.507 / - | 0.207 / - | 0.871 / - | 0.472 / - | 0.702 / - |
| ClonEx | 0.057 / 0.425 | 0.513 / 0.608 | 0.276 / 0.438 | 0.015 / 0.027 | 0.005 / 0.043 | 0.040 / 0.014 |
| ClonEx + CReLU | 0.196 / 0.325 | 0.558 / 0.610 | 0.213 / 0.275 | 0.039 / 0.029 | 0.067 / 0.003 | 0.012 / -0.014 |
| ClonEx+InFeR | 0.117 / 0.075 | 0.503 / 0.462 | 0.232 / 0.286 | 0.031 / 0.043 | 0.001 / -0.014 | 0.038 / 0.000 |
| R&D | **0.002 / -0.025** | **0.041 / -0.025** | **0.014 / -0.013** | 0.000 / 0.050 | 0.008 / 0.029 | 0.045 / 0.029 |

to be at a moderate level for the 'Random' sequence. It is worth mentioning that, as discussed in section 3, PPO exhibits a higher propensity for negative transfer compared to SAC.

In terms of forgetting, it appears that CRL methods, excluding ClonEx and R&D, also experience catastrophic forgetting. Given that SAC typically exhibits greater forgetting than PPO, one might infer that PPO is a more suitable choice for CRL. But this is not the case, as negative transfer rate of PPO is higher than that of SAC, resulting in a smaller number of trainable tasks in the sequence for PPO. Therefore, it is inappropriate to directly compare the forgetting of SAC and PPO.

In our previous findings, we observed that while the average success rate of ClonEx surpasses that of other CRL baselines, it still falls short of the average success rate achieved by R&D. However, the results indicate that ClonEx exhibits forgetting comparable to R&D. Hence, we can deduce that the performance degradation of ClonEx is attributed to negative transfer rather than forgetting.

## 5.4 THE EFFECT OF THE SIZE OF $\mathcal{M}_\tau$ AND $\mathcal{D}_\tau$

In this section, we additionally investigate the impact of the size of the expert and the replay buffer on the performance of R&D. To examine the effect, we conducted experiments by varying the size of the replay buffer used in the distillation phase from 10k to 1M, and the size of the expert buffer from 1k to 10k. We used 'Hard' sequence, which can be considered as the most challenging sequence in the previous experiments as it showed the highest negative transfer among the 3 sequences, and measured the average success rate of each task after all tasks were learned. Figure 6 illustrates the results. Note that $|\mathcal{D}_\tau|$ and $|\mathcal{M}^k|$ indicate the size of replay and expert buffer respectively. Both SAC and PPO algorithms show that as the replay buffer size increases, the average success rate also increase. This is because if the size of the replay buffer is too small, the total number of samples used for training the model decreases, leading to insufficient learning. When we varied the size of the expert buffer, we did not observe any noticeable differences. Based on this result, we can reduce the expert buffer size to achieve better memory efficiency.

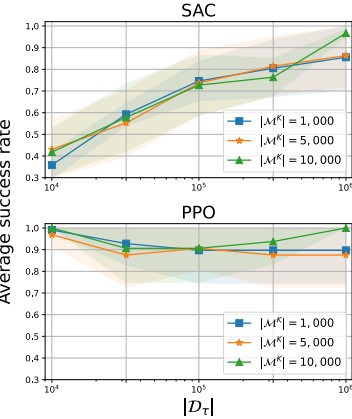

Figure 6: The average success rates of R&D with SAC and PPO on various $|\mathcal{D}_\tau|$ and $|\mathcal{M}^k|$.

## 6 CONCLUSION

In this paper, we first point out the catastrophic negative transfer which is an overlook problem in CRL community. By extensively explore the existence of negative transfer in environments from Meta-world, we figure that the negative transfer can occur quite frequently regardless of the RL algorithms. Based on these findings, we propose a simple yet effective baseline, dubbed as **R**eset and **D**istill (**R&D**), that can resolve both catastrophic negative transfer and forgetting, In the experiment, we carry out extensive experiments on long task sequences, and also conduct the analysis on R&D. For the future work, extending R&D to capturing the ability of forward transfer which accelerates learning new tasks and to vision signal-based RL can be considered.

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
