# 7 SUPPLEMENTARY MATERIALS

## 7.1 DETAILS ON 8 TASK GROUPS

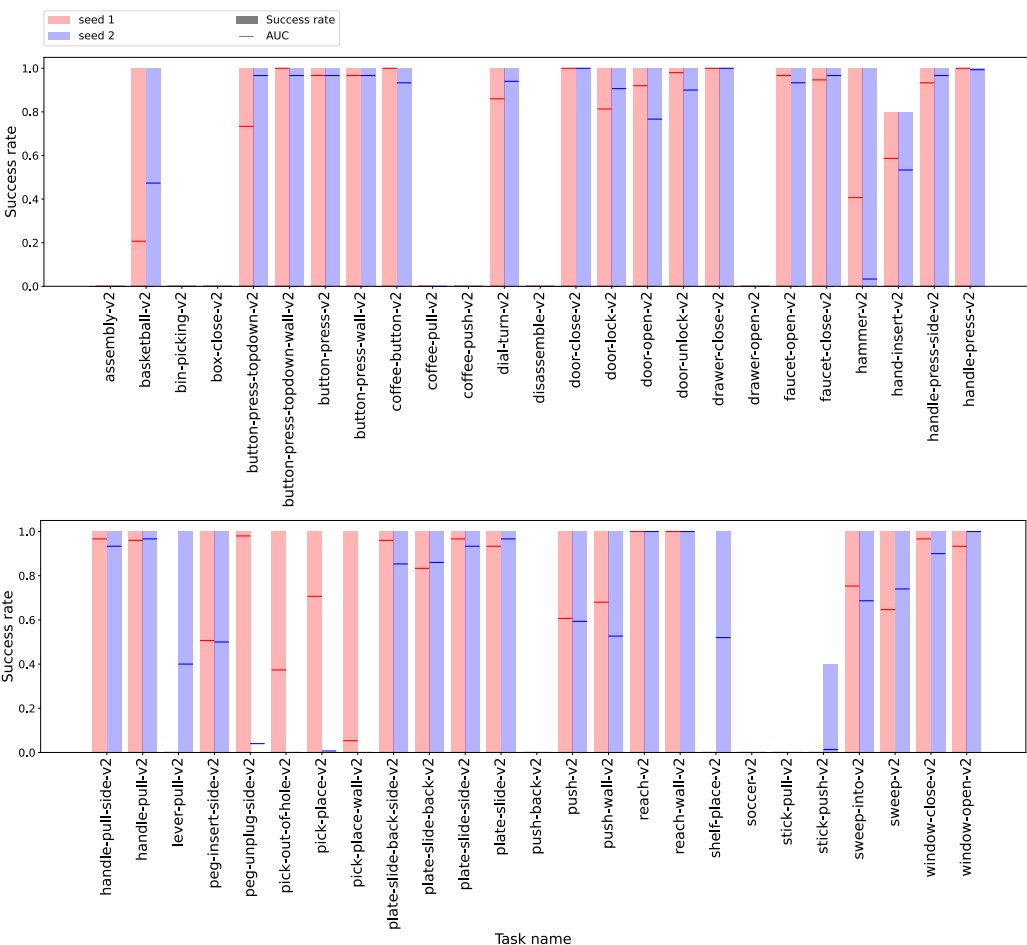

Figure 7: Success rates after training 50 task in Meta-World for 3M steps. SAC was used for training. Results from two different random seeds are distinguished by different colors. The bar plot represents the success rate, and the line marker represents the area under the curve (AUC) of the success rate curve obtained during training.

Prior to examining negative transfer in CRL, we identified tasks that could be learned within 3M steps among the 50 robotic manipulation tasks included in Meta-World Yu et al. (2020). Figure 7 illustrates the success rates when training the 50 tasks using the SAC algorithm Haarnoja et al. (2018) for 3M steps. In this figure, tasks with lower area under the curve (AUC) values can be interpreted as requiring a relatively larger number of steps for training. This implies that some tasks may not be learned within 3M steps in certain cases. Therefore, to identify negative transfer in specific tasks, it is necessary to prioritize tasks that can be fully learned within 3M steps, i.e., tasks with high success rates and AUC values. Following this criterion, we selected the following 24 tasks:

- `button-press-topdown`
- `button-press-topdown-wall`
- `button-press`
- `button-press-wall`
- `door-close`
- `door-lock`
- `door-open`
- `door-unlock`
- `faucet-open`
- `faucet-close`
- `handle-press-side`
- `handle-press`

- `handle-pull-side`
- `handle-pull`
- `plate-slide-back-side`
- `plate-slide-back`
- `plate-slide-side`
- `plate-slide`

- `push`
- `push-wall`
- `sweep-into`
- `sweep`
- `window-close`
- `window-open`

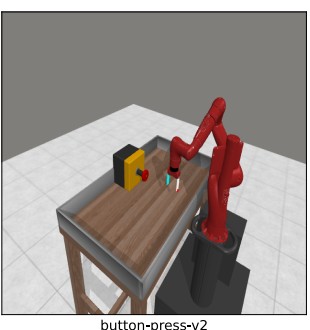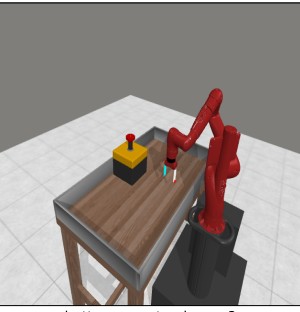

button-press-v2      button-press-topdown-v2

Figure 8: Visualization of `button-press` (left) and `button-press-topdown` (right).

As indicated by their names, the tasks can be classified based on similarity. For example, as seen in Figure 8, both `button-press` and `button-press-topdown` involve the robot pressing a button, with the only difference being the direction of the button. By grouping similar tasks together, the 24 selected tasks can be classified into a total of 8 groups.

- **Button**: `button-press-topdown`, `button-press-topdown-wall`, `button-press`, `button-press-wall`
- **Door**: `door-close`, `door-lock`, `door-open`, `door-unlock`
- **Faucet**: `faucet-open`, `faucet-close`
- **Handle**: `handle-press-side`, `handle-press`, `handle-pull-side`, `handle-pull`
- **Plate**: `plate-slide-back-side`, `plate-slide-back`, `plate-slide-side`, `plate-slide`
- **Push**: `push`, `push-wall`
- **Sweep**: `sweep-into`, `sweep`
- **Window**: `window-close`, `window-open`

## 7.2 TASK-WISE NEGATIVE TRANSFER

Figure 9 and Figure 10 show the two-task results with 13 tasks.

## 7.3 THE RESULTS WITH STATISTICAL SIGNIFICANCE

In this section, we report the statistical significance of the results shown in the Manuscript.

## 7.4 DETAILS ON EXPERIMENT SETTINGS

In the all experiments, we used Adam optimizer and the code implementations for all experiments are based on Garage proposed in Yu et al. (2020).

### 7.4.1 HYPERPARAMETERS FOR THE EXPERIMENTAL RESULTS

The hyperparameters for SAC and PPO are described in Table 3 and Table 4, respectively. For the hyperparameters on the CRL methods, the details are described as follows:

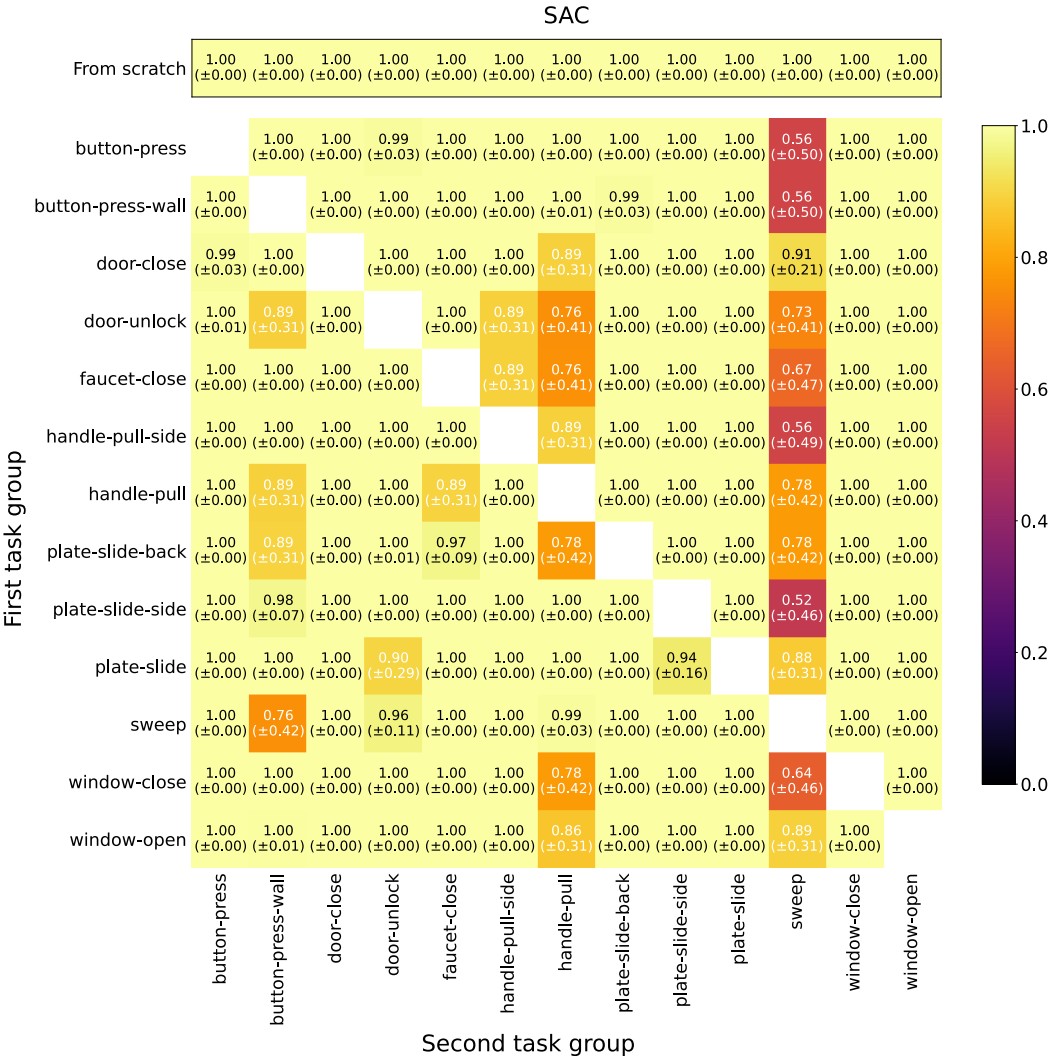

Figure 9: Task-wise negative transfer results of SAC on 13 tasks.

- EWC, P&C: The regularization coefficient was set to 1000

- BC: The regularization coefficient was set to 1, and the expert buffer size $|\mathcal{M}_k|$ was set to $10k$ for task $k$.

- R&D: The regularization coefficient was set to 1, and the expert buffer size $|\mathcal{M}_k|$ was set to $10k$ for task $k$. Furthermore, the replay buffer size $|\mathcal{D}|$ was set to $10^6$

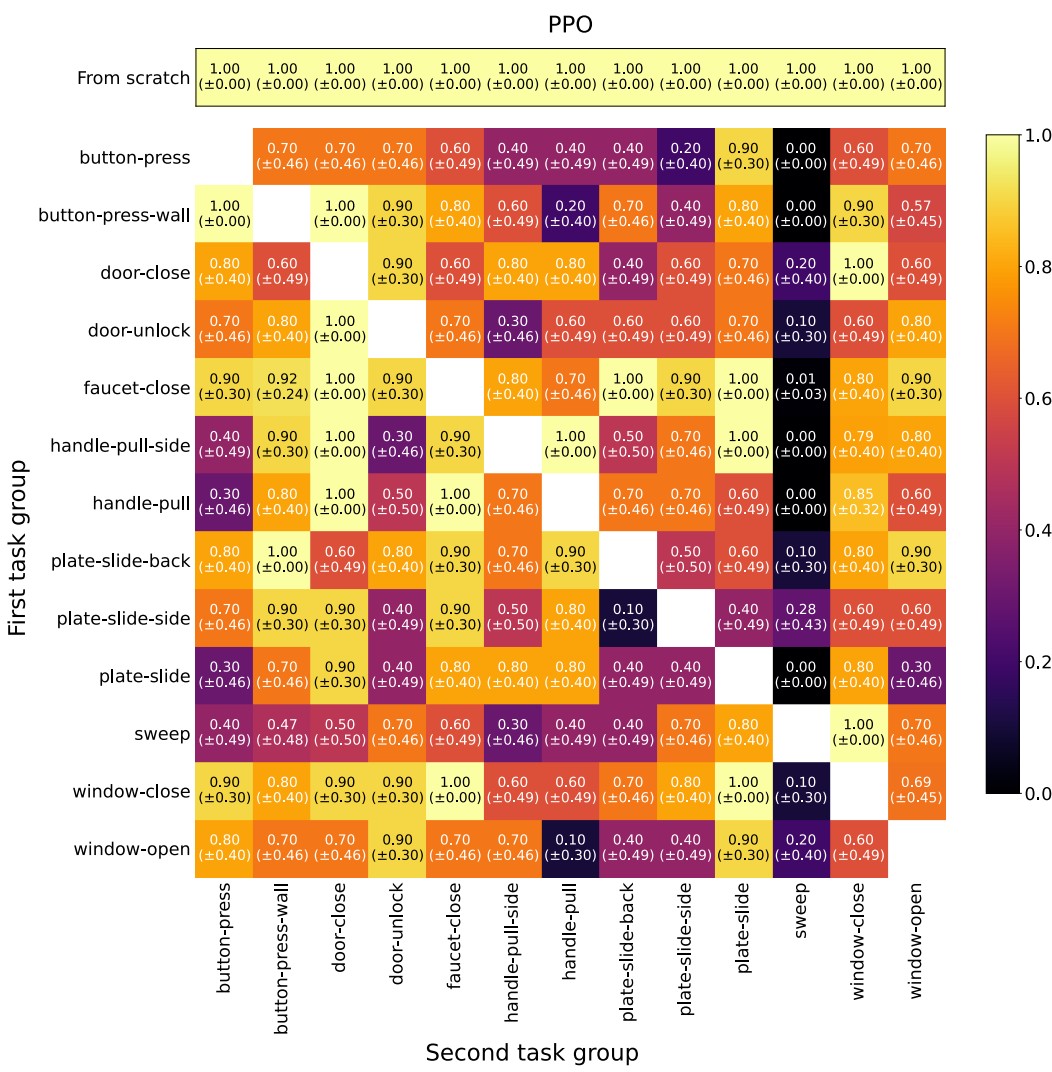

Figure 10: Task-wise negative transfer results of PPO on 13 tasks.

Table 2: The negative transfer and forgetting results with standard deviation. Note that the numbers after $\pm$ represent the standard deviation.

| Measure | Negative transfer | | | Forgetting | | |
|---|---|---|---|---|---|---|
| Sequence | Easy | Hard | Random | Easy | Hard | Random |
| | SAC | | | | | |
| Fine-tuning | $0.0955 \pm 0.0929$ | $0.5002 \pm 0.1236$ | $0.1925 \pm 0.132$ | $0.8997 \pm 0.0912$ | $0.5040 \pm 0.1333$ | $0.7766 \pm 0.1111$ |
| EWC | $0.0708 \pm 0.0813$ | $0.4567 \pm 0.0915$ | $0.2598 \pm 0.1294$ | $0.8517 \pm 0.1129$ | $0.5123 \pm 0.0969$ | $0.6714 \pm 0.1327$ |
| P&C | $0.0708 \pm 0.1134$ | $0.5065 \pm 0.1439$ | $0.2077 \pm 0.1517$ | $0.8714 \pm 0.1187$ | $0.4723 \pm 0.1338$ | $0.7023 \pm 0.1335$ |
| ClonEx | $0.0570 \pm 0.0768$ | $0.5130 \pm 0.1574$ | $0.2760 \pm 0.1322$ | $0.0146 \pm 0.0437$ | $0.0049 \pm 0.0632$ | $0.0397 \pm 0.0714$ |
| ClonEx + CReLU | $0.1958 \pm 0.1936$ | $0.5580 \pm 0.1166$ | $0.2132 \pm 0.1947$ | $0.0389 \pm 0.0557$ | $0.0671 \pm 0.0997$ | $0.0117 \pm 0.0291$ |
| ClonEx+InFeR | $0.1172 \pm 0.1030$ | $0.5032 \pm 0.1654$ | $0.2322 \pm 0.1655$ | $0.0311 \pm 0.0626$ | $0.0006 \pm 0.0666$ | $0.0377 \pm 0.1073$ |
| R&D | $0.0020 \pm 0.0232$ | $0.0412 \pm 0.0566$ | $0.0140 \pm 0.0603$ | $0.0000 \pm 0.0000$ | $0.0083 \pm 0.0359$ | $0.0454 \pm 0.0701$ |
| | PPO | | | | | |
| Fine-tuning | $0.3788 \pm 0.1866$ | $0.6238 \pm 0.1439$ | $0.4250 \pm 0.2318$ | $0.3614 \pm 0.1114$ | $0.3314 \pm 0.1117$ | $0.3357 \pm 0.1567$ |
| EWC | $0.5363 \pm 0.2493$ | $0.6763 \pm 0.1365$ | $0.3750 \pm 0.1250$ | $0.3186 \pm 0.1250$ | $0.2814 \pm 0.1591$ | $0.4300 \pm 0.0043$ |
| P&C | - | - | - | - | - | - |
| ClonEx | $0.4250 \pm 0.1785$ | $0.6075 \pm 0.1576$ | $0.4375 \pm 0.2183$ | $0.0271 \pm 0.0621$ | $0.0429 \pm 0.0655$ | $0.0143 \pm 0.0429$ |
| ClonEx + CReLU | $0.325 \pm 0.1392$ | $0.6100 \pm 0.1814$ | $0.2750 \pm 0.1458$ | $0.0286 \pm 0.0571$ | $0.0029 \pm 0.0086$ | $-0.0143 \pm 0.0769$ |
| ClonEx+InFeR | $0.0750 \pm 0.1696$ | $0.4625 \pm 0.2440$ | $0.2875 \pm 0.3115$ | $0.0429 \pm 0.0655$ | $-0.0143 \pm 0.0429$ | $0.0000 \pm 0.0000$ |
| R&D | $-0.0250 \pm 0.0500$ | $-0.0250 \pm 0.0500$ | $-0.0125 \pm 0.0375$ | $0.0500 \pm 0.0906$ | $0.0286 \pm 0.0571$ | $0.0286 \pm 0.0571$ |

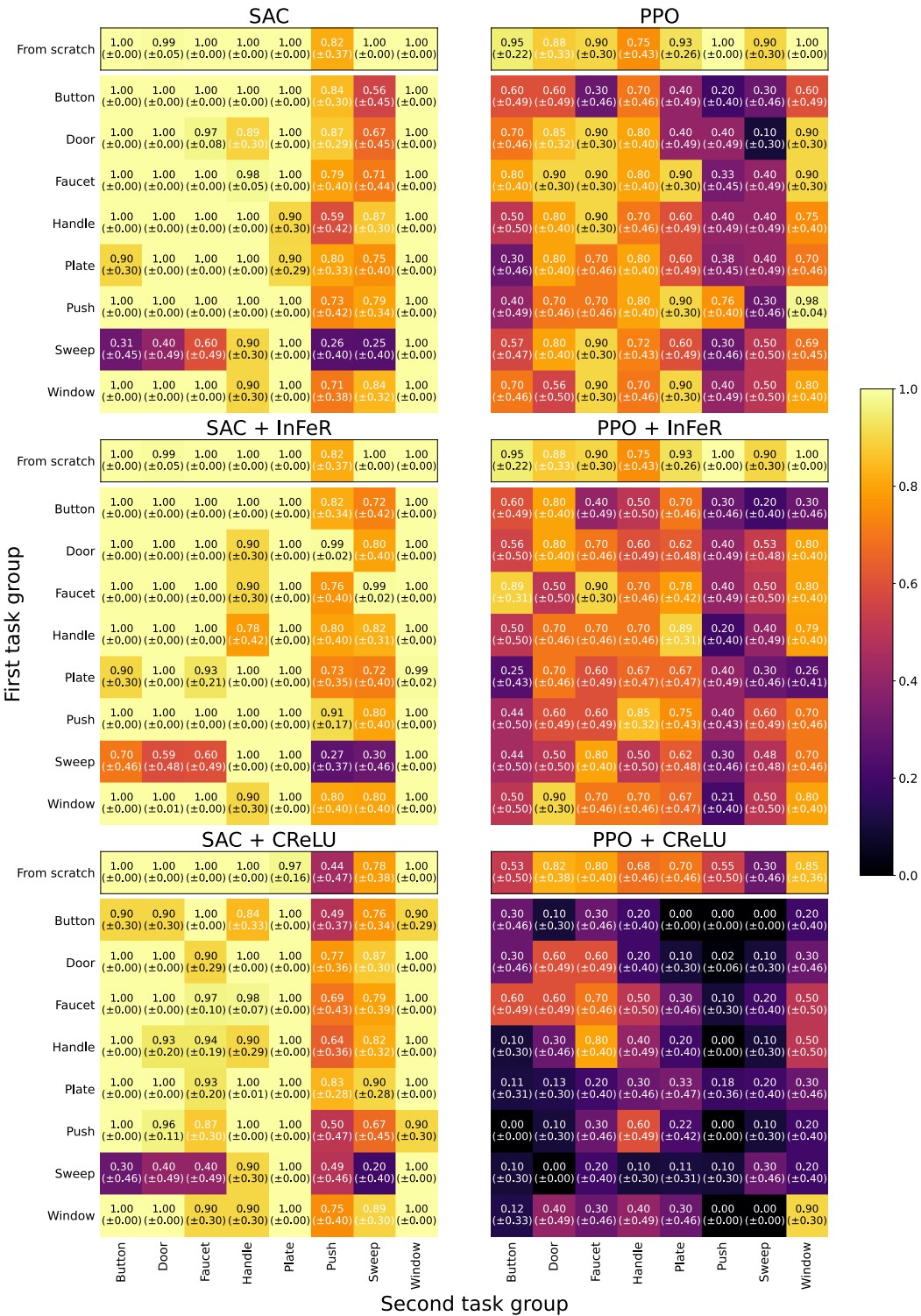

Figure 11: The results on training SAC and PPO with CReLU or InFeR. We include the standard deviation into the results proposed in the Manuscript.

Table 3: Model hyperparameters for SAC

| Description | Value | variable_name |
|---|---|---|
| Normal Hyperparameters | | |
| Batch size | 500 | batch_size |
| Number of epochs | 500 | n_epochs |
| Path length per roll-out | 500 | max_path_length |
| Discount factor | 0.99 | discount |
| Algorithm-Specific Hyperparameters | | |
| Hidden sizes | $(256, 256)$ | hidden_sizes |
| Activation function | ReLU | hidden_nonlinearity |
| Policy learning rate | $3 \times 10^{-4}$ | policy_lr |
| Q-function learning rate | $3 \times 10^{-4}$ | qf_lr |
| Mini batch size | 128 | buffer_batch_size |
| Replay buffer size | $10^6$ | capacity_in_transitions |
| Policy minimum standard deviation | $e^{-20}$ | min_std |
| Policy maximum standard deviation | $e^2$ | max_std |
| Gradient steps per epoch | 500 | gradient_steps_per_itr |
| Number of epoch cycles | 200 | epoch_cycles |
| Number of epochs | 30 | epochs |
| Soft target interpolation | $5 \times 10^{-3}$ | target_update_tau |
| Automatic entropy tuning | True | use_automatic_entropy_tuning |

Table 4: Model hyperparameters for PPO

| Description | Value | variable_name |
|---|---|---|
| Normal Hyperparameters | | |
| Batch size | 15000 | `batch_size` |
| Number of epochs | 200 | `epochs` |
| Path length per roll-out | 500 | `max_path_length` |
| Discount factor | 0.99 | `discount` |
| Algorithm-Specific Hyperparameters | | |
| Policy hidden sizes | $(128, 128)$ | `hidden_sizes` |
| Policy minimum standard deviation | 0.5 | `min_std` |
| Policy maximum standard deviation | 1.5 | `max_std` |
| Policy share standard deviation and mean network | True | `std_share_network` |
| Activation function | tanh | `hidden_nonlinearity` |
| Learning rate | $5 \times 10^{-4}$ | `learning_rate` |
| Likelihood ratio clip range | 0.2 | `lr_clip_range` |
| Advantage estimation | 0.95 | `gas_lambda` |
| Use layer normalization | False | `layer_normalization` |
| Entropy method | max | `entropy_method` |
| Loss function | surrogate clip | `pg_loss` |
| Maximum number of epochs for update | 32 | `max_epochs` |
| Mini batch size | 128 | `batch_size` |
| Value Function Hyperparameters | | |
| Value function hidden sizes | (128,128) | `hidden_sizes` |
| Activation function | tanh | `hidden_nonlinearity` |
| Initial value for standard deviation | 1 | `init_std` |
| Use trust region constraint | False | `use_trust_region` |
| Normalize inputs | Ture | `normalize_inputs` |
| Normalize outputs | True | `normalize_outputs` |

## 8 FIGURES FOR REBUTTAL

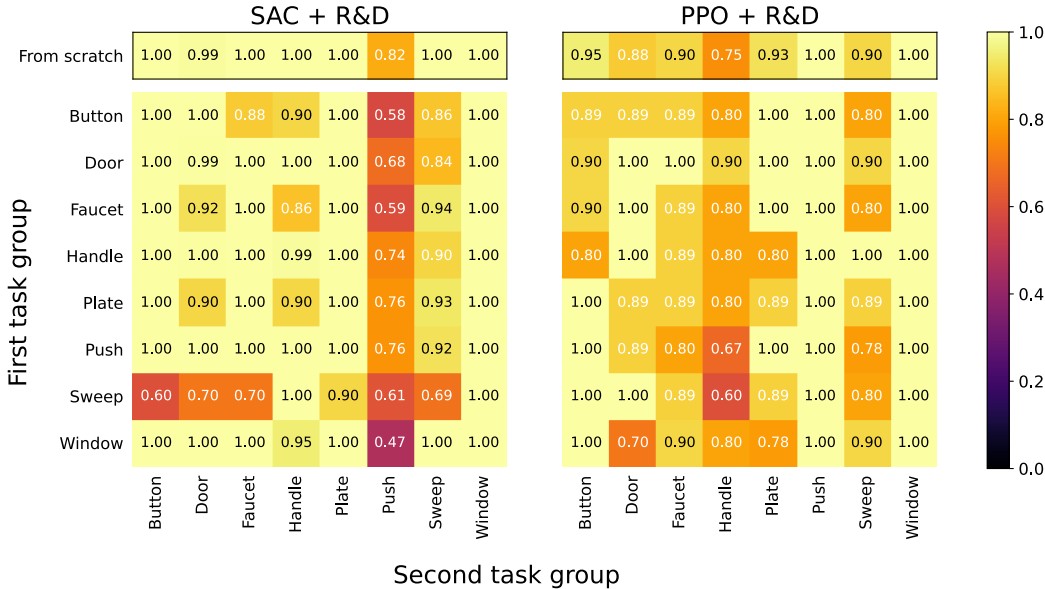

Figure 12: Negative transfer results on R&D.

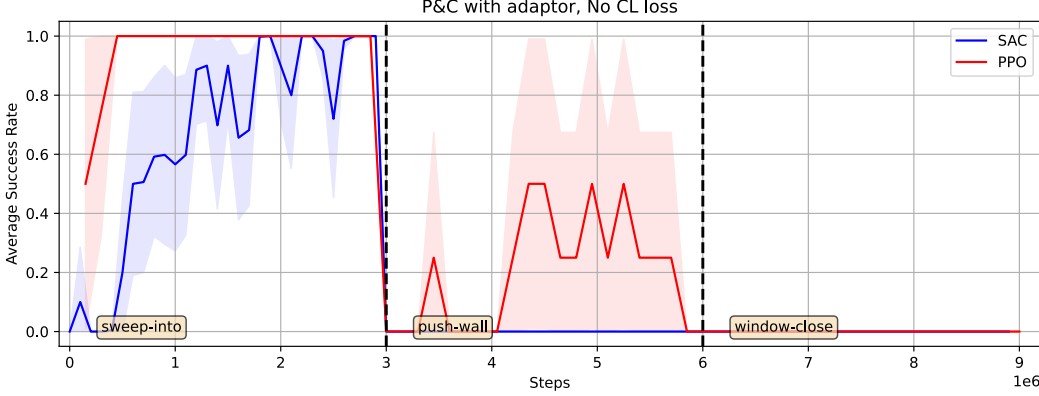

Figure 13: Success rates of SAC and PPO equipped with P&C in which the active columns are reset after learning task. Note that to show the negative transfer only, we removed the regularization loss at the compress step. For the results on each task learned from scratch, please refer to Figure 1 in the Manuscript.