# OpenReview forum: "Catastrophic Negative Transfer: An Overlooked Problem in Continual Reinforcement Learning"
_ICLR.cc/2024/Conference — Submitted to ICLR 2024_

### Official Review · Reviewer_p3gB · 2023-10-26

**Soundness:** 2 fair
**Presentation:** 3 good
**Contribution:** 2 fair
**Rating:** 3
**Confidence:** 5

**Summary:**

The submission studies the problem of negative transfer between tasks in a continual RL (CRL) setting. The authors posit that this issue is distinct from the plasticity/stability trade-off studied in the majority of the CRL literature. The manuscript contains an initial evaluation on pairs of Meta-World tasks that demonstrates that indeed learning a new task might be negatively affected because of the learning of a previous task. Then a new approach is proposed to simply re-initialize the network upon training each new task and then distill knowledge from the trained network into a shared network trained across all tasks. An evaluation on Meta-World tasks demonstrates that the approach avoids negative transfer.

**Strengths:**

######## Strengths ########
- The proposed method is clearly explained
- The evaluation on Meta-World demonstrates that the method is capable of learning complex tasks
- The empirical evaluation covers multiple baselines, sequences of tasks, and a few ablation studies

**Weaknesses:**

######## Weaknesses ########
- The problem of negative transfer between RL tasks has been well known for a long time -- though not as extensively explored, indeed
- The proposed solution is underwhelming -- in order to avoid negative transfer, the approach also rids itself of the possibility of achieving positive transfer
- The most closely related approach, Progress & Compress (P&C) was apparently misunderstood by the authors, leading to a misleading evaluation
- There is no distinct related work section

######## Recommendation ########

Unfortunately, I recommend that this manuscript is not accepted in its current form. While I agree with the authors that negative transfer has been understudied in continual RL, the phenomenon has been well known for a long time (e.g., [1]). A paper in this space should provide some novel insight or understanding of the problem, which I don't believe the current submission provides. The proposed solution, instead of seeking to avoid negative transfer while maintaining the possibility to achieve positive transfer, simply trains _without_ transfer, trivially sidestepping the problem (and the advantages of positive transfer). Critically, there is a fundamental flaw in the authors' understanding and implementation of P&C (and extensions to P&C), which I discuss in detail below. This error makes P&C unsuitable for avoiding negative transfer, which is in fact an issue that the P&C authors discussed in their original paper in 2018.

######## Arguments ########
- It is unclear what is the novelty of the setting studied in this work
    - The problem of negative transfer in RL is well known. Indeed, in the continual setting it has been somewhat tied with plasticity/capacity (probably incorrectly)
    - The authors' discussion of how it differs from these other issues is an interesting first step, but fails to provide any clear insight
    - What is the reason for this negative transfer? When does it happen? How can we avoid it?
    - There is a recent treatment of a very similar issue in [2] which the authors fail to reference in their work. In [2], the authors explicitly study one particular setting where negative transfer occurs: when two tasks require executing different actions in the same state.
    - It's unclear why the results of Figure 2 cannot be explained via loss of capacity/plasticity. While I do agree that that's probably not the cause (because finetuning doesn't restrict capacity or plasticity), I don't believe there's a clear enough argument in the result to clearly establish that claim.
- The solution doesn't really address the problem, but sidesteps it
    - The problem the authors are trying to address is that, when attempting to transfer knowledge from one task to the next, sometimes that leads to negative transfer
    - Instead of aiming to determine when this is the case or somehow prevent it from happening, the proposed solution simply _doesn't_ transfer any knowledge across tasks. Trivially, this solution avoids negative transfer -- but it also avoids positive transfer!
    - I encourage the authors to continue down this path, and seek to develop a solution that maintains the benefits of positive transfer when available, and avoids negative transfer when it would occur.
- There is a critical flaw in the authors' implementation of P&C
    - The authors claim that P&C is restricted to "one active column... without any re-initialization." Quoting from the original P&C paper: "Note that one could make this phase similar to naive finetuning of a network trained on previous tasks by not resetting the active column or adaptors upon the introduction of a new task. Empirically, we found that this can improve positive transfer when tasks are very similar. For more diverse tasks however, we recommend re-initialising these parameters, which can make learning more successful."
    - The first implication of this is that the authors' evaluation of P&C is incorrect.
    - The second, and perhaps more important, is that P&C already explicitly addressed the issue of negative transfer 5 years ago. Of course, P&C is not an end-all solution, but it clearly is a mechanism that seeks to avoid negative transfer, and they do this while still maintaining positive forward transfer.
    - Moreover, while indeed P&C originally didn't use replay, it is a trivial adaptation of P&C, which has already been successfully tried (Mendez et al., 2022).
- There is no distinct related work section. While the authors do a somewhat complete treatment scattered throughout the paper of continual learning and CRL approaches, they miss any mention to existing studies of negative transfer (e.g., [1,2]).

As one additional point, the choice of tasks that can be trained within 3 million steps is especially well suited for an algorithm that achieves no forward transfer -- it's trained only on the easy tasks, which we can learn by definition with no forward transfer.


[1] Taylor & Stone, "Transfer Learning for Reinforcement Learning Domains: A Survey." JMLR, 2009.

[2] Kessler et al., "Same State, Different Task: Continual Reinforcement Learning without Interference." AAAI, 2022.

**Questions:**

######## Additional feedback ########

The following points are provided as feedback to hopefully help better shape the submitted manuscript, but did not impact my recommendation in a major way.


Intro
- I question the claim toward the end of the intro that says that because the agent can still learn window-close, then it's not due to loss of capacity
    - First, are we just doing finetuning? Then why would there be a loss of capacity?
    - Second, maybe window close is more "compatible" with sweep-into and therefore doesn't require additional capacity -- it somehow "fits in the same space"
- I still think this roughly boils down to the idea of Kessler et al., where it's not possible for a single model to perform well on two tasks that require "opposite" behaviors given the same state. I hoped to see some discussion about this.

Sec 2
- The authors should clearly anticipate in the intro and in Sec 2.2 why behavior cloning is mentioned at all. When I got here I assumed it was to do something similar to Wolczyk et al., but was confused about why this wasn't mentioned before.

Sec 3
- The experimental setting for Figure 2 is not very clear.
    - Why are the first words supposed to denote whether tasks are similar? For example button press is a lot more similar to coffee button than to button press topdown wall
    - Are the 8 groups all size 3 or are they different sizes? Do they all have at least 2 tasks?
    - I think what confused me was that the authors stated that they used _all_ to refer to groups and not tasks, while I was expecting the evaluation to be over all pairs of _tasks_. Perhaps a bit of rephrasing could help make this clearer.
- How should we interpret the results over 13 tasks? Does this mean training 13 tasks in sequence? How are they chosen, why...?

Sec 5
- Figure 3/5 seems to only show the performance of the offline network. for R&D. If that's the case, then the curves should look like "steps" -- there's no reward improvement on during the training of each individual task, and then there's a jump at the end when the online model is distilled.
- Do Figure 3/5 show the performance averaged across all tasks so far or only the current task?
- The use of "long task sequence" throughout the paper to refer to sequences of 8 tasks is quite underwhelming.
- "To check that either CReLU or InFeR can be compatible to the CL baseline" -- what does this mean?

There is no related work section

Typos/style/grammar
- Final paragraph of Sec 1 -- section --> Section
- The citation format seems odd. Why is there a parenthesis within each parenthetical citation for the year?
- Sec 3, paragraph 2 -- single task --> single-task

---

> ### Author Response · Authors · 2023-11-23
> **Comments for Reviewer p3gB (Part I)**
>
> ### Weakness 1: The novelty of the negative transfer problem in CRL
>
> - As we already mentioned in the Introduction section, we agree that the negative transfer has been studied in a similar way like loss of plasticity or capacity loss. However, our experiments have shown that a proper solution for resolving the negative transfer in CRL setting has not been proposed in other works. Either CReLU or InFeR cannot solve this problem, and we stress that this problem cannot be explained via plasticity or capacity loss. Furthermore, as reviewer p3gB mentioned, this problem is not extensively explored in previous methods, and we think that we extensively explored the negative transfer problem in Meta-World environment.
> - For the more in-depth analysis about the distinction between negative transfer and capacity/plasticity loss, please refer to the global comment.
>
> ### Weakness 2: The proposed method just sidesteps the negative transfer problem
>
> As mentioned in the conclusion, we agree that our method is not capable of positive transfer. However, addressing negative transfer is a prerequisite before contemplating positive transfer. Despite previous studies, even in the case of P&C, attempting to achieve positive transfer, our experiments revealed that such efforts did not effectively tackle negative transfer. We have demonstrated that our method outperforms existing baselines by solely addressing negative transfer, without explicitly incorporating positive transfer in a straightforward manner. Hence, we assert that our contribution lies in this aspect.
>
>
> ### Weakness 3: There is a critical flaw in the implementation of P&C
>
> We acknowledge that we were not aware of the reference to a reset in P&C. We have further experimented with the implementation of P&C. These results are available for review in the Supplementary Materials of the revised manuscript. Our conclusion is that we cannot agree that P&C solves the negative transfer issue. In our experimental setup, we implemented the algorithm proposed by P&C as it is, attaching an adaptor to the policy and receiving features from the knowledge base policy(`policy_kb`). When learning a new task, we reset the policy and the Q function (or the value function), and proceeded with learning again. Note that when learning second task, the adaptor is trained from randomly initialized network, and the third task is learned with pre-trained adaptor from the second task. In the result, we can see that the task is not learned at all. We think this is due to the part where the information from `policy_kb` is passed through the adaptor. It might seem that applying resetting to P&C would solve the negative transfer, but in our experiments, we found that it did not solve the negative transfer at all. Therefore, we can see that P&C is not able to maintain positive transfers and is also adversely affected by negative transfers.
>
>
> ### Weakness 4: There is no distinct Related Work section
>
> Our intention was to have the Introduction section serve as an introduction to the related work, but this seems to have been confusing. We will add a new Related Work section to describe more about works such as transfer learning later on.
>
>
> ### Weakness 5: The choice of tasks that can be trained within 3 million steps is well suited only for an algorithm that achieves no forward transfer
>
> We focused our experiments on the occurrence and resolution of negative transfers. We excluded tasks that were not learned within 3 minutes of the CRL because they would not allow us to see the extent of negative transfer. Though it would be better to consider more hard and complex tasks to show the forward transfer in our experiment, we think that resolving the negative transfer is our major challenge, and achieving the forward transfer remained as future work.

---

> > ### Author Response · Authors · 2023-11-23
> > **Comments for Reviewer p3gB (Part II)**
> >
> > ### Question 1: Questions regarding the Introduction
> >
> > - Firstly, the results in Figure 1 are from conducting only finetuning. We are highlighting the issue of negative transfer persisting even in the simple finetuning scenarios. Similar concerns were previously raised in the context of capacity/plasticity loss.
> > - Secondly, I think what you're saying is somewhat similar to what we're trying to say. This is because as the agent learns, it gets better at some tasks and worse at others as it fits into one task.  In this case, from the network's point of view, the only difference from learning from scratch is the parameter initialization. This is the reason we directed our attention towards the re-initialization of the network.
> >
> > ### Question 2: Our work roughly boils down to the idea of [1]
> >
> > - [1] discusses the difficulty of learning two tasks with the same state space and opposing behaviors. While there may be some similarity in addressing dissimilarity between tasks, there are significant differences in our work.
> > - To simplify, let's consider the scenario of sequentially learning two tasks, A and B, with the same state space and opposing target behaviors in the order of A→B. [1] does not address the situation where learning B is hindered by A. Instead, it focuses on situations where, as B is well-learned, A is forgotten since it requires opposite actions. In contrast, the negative transfer proposed in our work deals with a scenario where learning B becomes impossible due to prior learning of A. This is not simply because A and B target opposite behaviors; our experiments demonstrated the inability to learn B even when all information about A was lost during fine-tuning without any regularization.
> > - Such a problem cannot be resolved by methods like OWL mentioned in [1]. OWL, in brief, attaches heads corresponding to each task to the network and applies EWC or other continual learning methods to the shared parameters. In fact, all of our experiments use multi-head agents, but negative transfer is still not solved.
> > - Additionally, interference due to opposite behavior, as mentioned in [1], intuitively seems plausible but somewhat contradicts our experimental results. For example, when sequentially learning 'window-close' and 'window-open,' which have the exactly same states and opposite goals, we observed successful learning without negative transfer.
> >
> > ### Question 3: Why Behavioral Cloning is mentioned in Section 2?
> >
> > Since we mainly use the Behavioral Cloning (BC) proposed in [2] to prevent forgetting in our method, we explain the details on BC as a preliminary work. If it is awkward to metion BC in this section, we will write additional brief introduction of BC in the Related Work section.
> >
> > ### Question 4: Why are the first words supposed to denote whether tasks are similar?
> >
> > We determined the similarity of the task based on the similarity of the visualization of the states and the task characteristic rather than the similarity of the action. For example, the tasks in 'faucet' group are 'faucet-close' and 'faucet-open'. The type of those tasks is just turn the 'facet' to the left or right. Though the 'button-press' and the 'coffee-button' may be similar, they just share the **actions**, not the overall type of tasks and the states.
> >
> >
> > ### Question 5: Questions about task groups
> >
> > - First, the number of tasks in each group varies across all task groups, and all task groups contain at least 2 tasks. We will clarify this part in our camera-ready version.
> > - For the results on the 13 tasks, we only trained two tasks sequentially, and our intention was to visualize the negative transfer results between two individual tasks, not the task groups. Furthermore, the interpretation on this results is not different from the group-wise results. Those resutls are just task-wise view of the negative transfer.

---

> > > ### Author Response · Authors · 2023-11-23
> > > **Comments for Reviewer p3gB (Part III)**
> > >
> > > ### Question 6: Questions about Figure 3 and 5
> > > - To begin with, the results depicted in Figure 3 and 5 represent the averaged success rate across all 8 tasks.
> > > - As you understood, the average success rate of R&D provided in Figures 3 and 5 represents the performance of the offline actor. It is true that the offline actor remains unchanged during the 3M steps in which the online actors are trained. However, if the success rate of the offline actor is represented as a step function according to the environment frames as you said, there is room for interpretation that the offline actor is used in the process of learning individual tasks. That is why we put markers at each point where the offline actor is trained and connected them with lines for visibility.
> > > - Furthermore, even though the offline actor is not trained during the process of learning individual tasks, this does not mean that there is no information gained while training the online actor. If we train the online actor for only 1M steps, which is 1/3 of the original training time, and then distill it to the offline actor, there would be a corresponding performance improvement to 1M. Therefore, although the training of the offline actor occurs in a discrete manner, we believe that the improvement in performance can also be interpreted as somewhat continuous.
> > >
> > > ### Question 7: The use of "long task sequence" refer to sequences of 8 tasks is underwhelming.
> > >
> > > Our intention regarding the "long sequence" is to present results for a greater number of tasks than discussed in the previous chapter, where we covered experiments with two tasks. While the sequences of length 8 may fall short of what you consider sufficient, we believe it was adequate to showcase the results we aim to provide.
> > >
> > > ### Question 8: "To check that either CReLU or InFeR ..." - what does this mean?
> > >
> > > The meaning behind this comment is to show whether the CL methods equipped with CReLU or InFeR can resolve both catastrohpic negative transfer and forgetting in the long sequence experiment. If it is confusing, we revise this comment more clearly.
> > >
> > >
> > > [1] Kessler et. al., "Same State, Different Task: Continual Reinforcement Learning without Interference", AAAI, 2022.
> > >
> > > [2] Wołczyk et. al., "Disentangling Transfer in Continual Reinforcement Learning", NeurIPS, 2022

---

### Official Review · Reviewer_oDoz · 2023-10-27

**Soundness:** 4 excellent
**Presentation:** 4 excellent
**Contribution:** 4 excellent
**Rating:** 8
**Confidence:** 4

**Summary:**

This paper focuses on continual reinforcement learning (CRL) and negative transfer. They show using Meta-World RL environments where the negative transfer occurs. For negative transfer, they delve into severe plasticity degradation depending on the learned task sequence. With this negative transfer phenomenon they propose an effective method denoted as Reset & Distill (R&D) thereby mitigating negative transfer and forgetting. For negative transfer they use MetaWorld and utilize SAC and PPO to show the negative transfer scenario in 8 task groups. In Table 1, they showcase how their method R&D has the lowest negative transfer and forgetting while comparing among fine-tuning, EWC, P&C, ClonEx, ClonEx+CReLU, and ClonEx+InFeR.

**Strengths:**

Great amount of experiments and a good amount of baselines to show why your method performs better than the others. In addition, an interesting set of experiments especially with the longer sequence. Plus you have three different settings: easy, hard, and random to simulate different varying conditions.

The introduction is quite motivating and makes it quite easy to understand the motivation. Plus you provide a plethora of previous works and mention what you are particularly working on in terms of negative transfer. For Figure 1, that is a good example to show how the negative transfer happens and makes it very easy to understand why you are wanting to work on this problem, much kudos.

**Weaknesses:**

Experiments:
For the experiments with CReLU or InFeR, why is R&D not compared, it would help bolster your method if in both scenarios R&D shows benefit. If not, it would be helpful to state why R&D is not put here. I can understand from logic, you state that it is not helpful. Yet, you used R&D, in a separate application, so it feels like there is a missing component since you used R&D in a different application.

Writing & Visualization:
For Figures 3 and 5, it may benefit to have each SAC and PPO separate. I understand that the paper limit is a hard problem. Consider in the supplementary material, to have each SAC and PPO plots in there so it makes it easier to see the separation.

For some of the captions like Table 1 or Figure 6, you state the information but it would be helpful to provide a conclusion of the results so it can spell to the reader what the message of the figure or table is.

Also in the second paragraph you mention that you ran 10 different random seeds,  how is that different from Figure 2 where you mentioned 3 seeds? May you please provide more details into it?

In 7.3 you mention statistical significance but where are the p-values?

**Questions:**

Please refer to the weaknesses section.

---

> ### Author Response · Authors · 2023-11-23
> **Comments for Reviewer oDoz**
>
> ### Weakness 1: Why is R&D not compared with CReLU or InFeR?
>
> Thank you for the valuable suggestion. We included the results from R&D in the Supplementary Materials. In this result, R&D effectively resolves the negatvie transfer. Though some tasks (e.g. the tasks after learning the tasks in 'Sweep group') are still hard to be learned in SAC, now the tasks in 'Sweep' group can be learned when those tasks lie in the second task. Furthermore, many tasks do not suffer from negative transfer in PPO.
>
> ### Weakness 2: Suggestions about writing and visualization
>
> - Firstly, apologies for the confusion caused by the typo. We also used 10 random seeds in the experiments for Figure 2.
> - We will take into account the other suggestions later on. Thank you for the feedback.
>
> ### Weakness 3: Why has the p-value not been provided?
>
> - While we did not directly provide p-values in the Supplementary Materials, we have provided enough information through standard deviation for some inference.
> - It seems that the term "statistical significance" may have caused some misunderstanding. We will make corrections to clarify this.

---

### Official Review · Reviewer_u4tG · 2023-10-31

**Soundness:** 3 good
**Presentation:** 3 good
**Contribution:** 3 good
**Rating:** 5
**Confidence:** 4

**Summary:**

In this paper, they address the issue of negative transfer in continual reinforcement learning by adding reset and distill strategies. They use the resetting strategy for online actor-critic and use distillation for the offline actor. They show on Meta-world tasks that these strategies mitigate both catastrophic negative transfer and forgetting in CRL.

**Strengths:**

- The papers is well written and organized.

**Weaknesses:**

- The innovation is not enough as a contribution.
- Other works show that resetting works for continual learning. It is not new to add Resetting and KL divergence to the loss function to mitigate forgetting.
- In the section 5.1, it compares fig 2 with fig 4 roughly. However it is not enough for the claim. It is needed to compare precisely the numbers in the two figures.
- There is not evidence or proof for remark 1 to say the reason is reward signal. There are different reasons for loss of plasticity.
- How is the offline actor trained?

**Questions:**

- What is the beneficial use of R&D vs just training each task from scratch like just doing resetting?
- The results in table 1 are very close together. It is not sufficient to compare just the numbers. It needs to clarify if they are significantly different.
- What is exactly the expert buffer?
- In the baselines, it would be worth to compare with just resetting agent or training an agent from scratch.
- What is the evidence for Remark 1? any proof?

---

> ### Author Response · Authors · 2023-11-23
> **Comments for Reviewer u4tG**
>
> ### Weakness 1: The innovation is not enough and other works show that resetting works for continual learning
>
> We appreciate the comments and would like to respond; however, we are facing challenges due to the limited availability of specific details. As far as we know, ClonEx introduced behavioral cloning (KL divergence) to prevent forgetting, but it does not address negative transfers. If there are other works with parameter resetting and KD that we missed, we would really appreciate it if you could let us know.
>
> ### Weakness 2: Figure 2 and 4 should be compared precisely
>
> We believe you had this question based on the following statement in Section 5.2 of our manuscript.
>
> > The findings reveal that fine-tuning with CReLU and InFeR yields similar success rates when compared to the results presented in Figure 2.
>
> What we wanted to emphasize with Figure 4 is that the solutions designed to address traditional capacity/plasticity loss, such as InFeR and CReLU, do not address negative transfer. For this purpose, we believed it was more important to compare the results with learning a single task from scratch rather than with finetuning (Figure 2). We have recognized that the wording in the manuscript could be potentially misleading, so we will fix it.
>
>
> ### Weakness 3: There is no evidence or proof for Remark 1
>
> For the response to Remark 1, please refer to the global comments.
>
> ### Weakness 4: How is the offline actor trained?
>
> First, a randomly initialized network (online actor) is used to learn each task. We use a known RL algorithm (SAC and PPO in our case) for training. In the process of training the online actor, there will be a replay buffer stored. If you use an on-policy algorithm such as PPO, you need to save the replay buffer after training. The states and actions in this stored replay buffer contain the knowledge of the online actor. By distilling this knowledge to the offline actor, the offline actor learns.
>
> ### Question 1: What is the beneficial use of R&D compared to just training each task from scratch?
>
> If we train each task with a re-initialized network, it will forget everything learned in the previous tasks, even though it can learn the current task well. Continual learning is not just about effectively learning the current task; it is crucial not to forget the content of previously learned tasks. Simple resetting alone cannot preserve performance on tasks learned earlier, so additional mechanisms are necessary. We solved the forgetting problem by adding behavioral cloning of the previous tasks to the knowledge distillation for the current task.
>
> ### Question 2: The results of Table 1 are very close and it is not sufficient to compare just the numbers
>
> Additional explanations about the interpretation of the results of Table 1 have been provided in the global comments. So please refer to them for further clarification.
>
> ### Question 3: What is exactly the expert buffer?
>
> Let's consider learning tasks A→B sequentially. The policy immediately after learning task A can be seen as an expert for task A. To retain the learned information at this point, behavioral cloning in continual learning involves storing the actions of the expert for states in task A in a buffer. Later, when learning task B, regularization is performed using the state-action pairs in this buffer. In summary, the buffer containing state-action pairs for the previously learned task is referred to as the expert buffer.
>
> ### Quenstion 4: In the baselines, it would be worth to compare with just resetting agent or training an agent from scratch.
>
> As mentioned earlier, simply resetting the agent for each task is not effective in continual learning. While this method may lead to effective learning of the current task, it results in a significant loss of information for previously learned tasks, ultimately leading to very poor performance.
>
> ### Question 5: What is the evidence for Remark 1?
>
> Please refer to the global comment for a response to Remark 1.

---

### Official Review · Reviewer_274b · 2023-11-04

**Soundness:** 2 fair
**Presentation:** 3 good
**Contribution:** 1 poor
**Rating:** 3
**Confidence:** 3

**Summary:**

This paper investigated a new empirical phenomenon in continual RL, dubbed catastrophic negative transfer. Particularly, algorithms tend to fail (lack plasticity) in the new task initialized from the previous task, even without incorporating the techniques to mitigate catastrophic forgetting. Next, the authors proposed a new strategy called Reset and Distill highly based on the behavior cloning approach to eliminate the catastrophic negative transfer phenomenon.

**Strengths:**

* The paper is well-organized, well-written and easy to follow.

* The experiments are extensive given the considered algorithms and environments.

**Weaknesses:**

* The conclusion is skeptical and not reliable, which needs further investigation.

* The proposed algorithm is straightforward and lacks technical contribution, although it is effective and technically sound in my opinion.

**Questions:**

The authors claimed a new phenomenon is overlooked in continual RL, but personally, the resulting conclusion is counterintuitive, and skeptical and may not be reliable without further investigations. Here are my explanations.

* Based on my knowledge in Q learning in the tabular setting, the convergence of Q learning (related to theoretical parts in certain stochastic processes and approximations) does not rely on initialization. Please refer to the well-accepted proof in [1], especially Theorem 1, although it is only a manuscript. This implies at least in the tabular setting, even with a very bad initialization, Q learning can always converge to an optimal one in the new environment under mild conditions. This result conflicts with the current manuscript, although this paper targets on deep RL setting. Thus, it is too early to conclude that catastrophic negative transfer commonly exists, and the reason that happens especially in deep RL settings should be carefully investigated and discussed. However, this paper has failed to do that. It would be more reasonable for me if the conclusion is this phenomenon is very unlikely to happen in tabular settings, while it is prone to occur in deep RL cases as the optimization/plasticity is hard in the current task under many initializations from the previous tasks. Therefore, I am not convinced by the current conclusions and strongly suggest more investigation and discussion about this phenomenon.

* The reason behind this phenomenon is also not convincing to me. This paper hypothesizes that the reward in the current tasks is not sufficient to guarantee convergence. Although I mostly agree with the claim, personally the deeper reasons may be the initializations from the previous tasks have a detrimental effect on the current tasks, especially in deep RL settings. Another hypothesis or potential concern is the training time is not sufficient, or the algorithm is not advanced enough to tolerate different initializations. For example, although SAC and PPO are commonly used, they may not be advanced enough to converge very well in these environments. I am thus skeptical of the conclusion. I even think some acceleration techniques used in RL algorithms can also help to mitigate this phenomenon, and thus the issue this paper studies may not be critical.


* The proposed algorithm is too straightforward. In my opinion, this resulting algorithm, although I agree it is well-motivated, effective and technically sound, is highly based on behavior cloning. It seems that it is equivalent to behavior cloning that additionally considers the memory buffer in the current task. The deficiency is also apparent as it can almost double the training costs (need to train an extra policy). As such, the proposed algorithm lacks technical contribution.

### More Detailed questions and comments

* What is the difference between the catastrophic negative transfer and plasticity? In my opinion, they are almost the same.

* Although Metaworld is commonly used, the empirical study on this environment may not generalize to broader environments. Also, the results of PPO and SAC cannot be generalized to all RL algorithms as claimed in this paper.

* In Figure 4, why PPO+CReLu performs even worse than PPO in the two-task continual learning? Why CReLu is not useful here?

* How many seeds are used in Table 1? It seems R&D is competitive with ClonEX without significant improvement.

* The considered baselines are limited and the straightforwardly using behavior cloning in the proposed algorithm may not be superior to other state-of-the-art continual RL baselines.

Overall, although this paper studies an overlooked phenomenon and did extensive experiments, empirical results tend to be fragile and may not generalize to broader settings. Some conclusions may be misleading and should be further investigated in the future before made faithfully.

[1] http://users.isr.ist.utl.pt/~mtjspaan/readingGroup/ProofQlearning.pdf

---

> ### Author Response · Authors · 2023-11-23
> **Comments for Reviewer 274b (Part I)**
>
> ### Question 1: Convergence of Q learning
>
> - First, there is a difference between tabular setting and deep RL initialization. While tabular initializes the output of the Q function directly, initialization of the Q network in deep RL means initialization of the network parameters.
> - This difference is even more pronounced for Q function updates. In the tabular setting, we directly change the output of the Q function according to the Bellman Equation, but we define a loss function and update the Q function by gradient descent in deep RL.
> - As far as we know, convergence in deep RL is not as well understood as in tabular settings. Therefore, a direct comparison between tabular RL and deep RL is somewhat inappropriate. For this reason, we did not address the theoretical convergence in deep RL. However, we have shown through extensive experiments that learning in deep RL can be affected by initialization, and it is not simply due to the loss of plasticity or capacity of the network.
>
> ### Question 2: Question about Remark 1
>
> - Please refer to the global comment for a response to Remark 1.
> - We also think that training time can be an important point. In many works, such as [1], the number of frames used per task in the Meta-World is set to 1M. In order to exclude the situation of running out of frames as much as possible, we set the number of frames per task to 3M, and then conducted the experiment by selecting only the tasks that can be learned well within 3M from scratch with 10 random seeds. Given that these tasks are effectively trained within 3M steps using well-known RL algorithms such as SAC or PPO, it is difficult for us to regard this as an issue.
> - Regarding the acceleration technique, we first thought that the policy trained on the previous task is not good at exploration, so we tried to train it by adding an auxiliary loss that forces it to some extent. However, this method cannot solve negative transfer.
> - Also, the ClonEx[1] that we adopted as a baseline not only utilizes behavioral cloning to prevent forgetting but also applies several exploration techniques to enhance learning efficiency. However, as evident from our experimental results, these methods did not effectively address negative transfer.
> - Thus, in our opinion, currently known acceleration techniques are not designed with negative transfer in mind, so it is unlikely that they can fundamentally solve it.
>
>
> ### Question 3: The proposed algorithm is too straightforward and is highly based on behavioral cloning
>
> The main contribution of our method is to reset the network before learning each task to avoid negative transfer. We also applied knowledge distillation to transfer this information to the continuous learner. It may seem similar to behavioral cloning, but it is only used as a means to an end. What we want to emphasize is that even this simple method can effectively alleviate negative transfer. Also, although our method takes two processes to learn the current task, it is not a big problem because the process of distilling the knowledge of online actors (learners from scratch) to offline actor (continual learner) requires very little time compared to learning online actors.
>
> ### Question 4: What is the difference between the catastrophic negative transfer and plasticity?
>
> We have presented a comprehensive analysis of the distinctions between negative transfer and capacity/plasticity loss. Your reference to this analysis would be greatly appreciated.
>
> ### Question 5: The empirical study on this environment may not generalize to broader environment
>
> We could have drawn even stronger conclusions if we had conducted experiments on different environments. However, MetaWorld itself utilizes a substantial number of tasks, and notably, all tasks share the same state space and actions (although the distribution of states may vary). We chose to conduct experiments in MetaWorld because we believed that these characteristics most clearly highlight the issue of negative transfer that we aim to address in continual RL.
>
> ### Question 6: Why CReLU is not useful in Figure 4?
>
> Through our experiments, we confirmed that learning a single task becomes challenging when applying CReLU. You can observe this from the results labeled 'From Scratch'. We believe that the observed performance degradation is a result of the difficulty in learning the single tasks. While the exact aspects causing these differences have not been identified, exploring them falls outside the scope of our investigation.
>
> ### Question 7: It seems R&D is competitive with ClonEx without significant improvement
>
> The results in Table 1 were negative transfer and forgetting measures computed based on the results shown in Figure 3 and 5. Therefore, a total of 10 random seeds were employed. Additional explanations regarding the interpretation of the results have been provided in the global comments; please refer to them for further clarification.

---

> > ### Author Response · Authors · 2023-11-23
> > **Comments for Reviewer 274b (Part II)**
> >
> > ### Question 8: The considered baselines are limited and behavioral cloning is not superior to other state-of-the-art continual RL baselines
> >
> >
> > - Existing continual RL works primarily focus on addressing catastrophic forgetting. To the best of our knowledge, ClonEx[1] is currently one of the best performing continual RL algorithms, and it effectively addresses the forgetting problem, as shown in our experimental results. Forgetting is minimal in the results when ClonEx is applied, making the need for additional baselines to compare forgetting unnecessary.
> > - Furthermore, ClonEx considers not only behavioral cloning but also incorporates several exploration techniques to account for forward transfer. However, we deemed this alone insufficient as a baseline for addressing negative transfer. Therefore, we augmented ClonEx with solutions proposed in InFeR and CReLU, addressing capacity/plasticity loss, as additional baselines. While InFeR and CReLU contribute to improving ClonEx's performance to some extent, as observed in Figure 5 and Table 1, they do not provide a fundamental solution.
> > - As mentioned earlier, the most significant contribution of our method is the reset of parameters before learning each task to avoid negative transfer. Our method does not simply apply behavioral cloning, and it can effectively address both forgetting and negative transfer.
> >
> >
> > [1] Wolczyk et. al., "Disentangling transfer in continual reinforcement learning", NeurIPS, 2022.

---

### Author Response · Authors · 2023-11-23
**Global comments for all reviewers**

We deeply apologize for the delayed response to your reviews, which regrettably occurred due to personal reasons. Your understanding in this matter would be greatly appreciated.

### Regarding Remark 1

> **Remark 1.** The rationale for our method is that since the reward signals are not strong enough to always adapt the pre-trained actor and critic networks to the current task …
- We are not suggesting that there is something wrong with the reward function itself. In a traditional supervised setting, the effects of negative transfer are not as prominent as in RL. The main difference between the two is that a clear true target is used in supervised setting, whereas a target based on the reward signal is utilized in RL. Therefore, our hypothesis is that the reward signal in reinforcement learning is not sufficiently potent to rectify inadequate initialization.
- Although we intended this to be the case, we agree that the statement above is misleading. We will modify it.

### Regarding the interpretation of Table 1

There were concerns raised about the absence of a significant difference in the results between R&D and ClonEx in Table 1. We, however, disagree that the results of R&D are merely on par with the other baselines. Intuitively, the negative transfer measure we defined can be thought of as the approximate percentage of tasks in which a negative transfer occurred out of all 8 tasks. While this is less pronounced in the Easy sequence, in the Hard sequence even the best performing baseline has a value above 0.5. This means that more than half of the tasks in the entire sequence are not trained at all. Our method, on the other hand, has values close to zero regardless of the sequence, and even some negative values (although we believe this is due to chance). As you can see from Figure 3, our method learns all tasks in the sequence well.

### Regarding the difference between the negative transfer and capacity/plasticity loss

- In case of the results of Figure 1 in our manuscript, we clarify the interpretation on the difference between the negative transfer and plasticity / capacity loss. First, in [1], when we fine-tune the value (or Q function) network, the network gradually loses its capacity on fitting new targets. And as a result, it cannot fit the reward signals given from the environments. In Figure 1, the success rates of second task are 0 during the whole training steps, and based on the interpretation on the capacity loss, all remaining capacity in the value network has been depleted. Therefore, we expect that it is hard to learn the following tasks because of the depleted capacity. However, unlike our expectation, the agent can learn the third task. We think that this phenomenon cannot be explained through the capacity loss. If the capacity has not been depleted, the agent can learn the new task, and vice versa. However, our results are not belonging to both cases. In case of the primacy bias [2] which is highly similar to the capacity loss, if we assume that the agent is highly biased toward the data collected from first task, it cannot learn all the following tasks. However, since the third task can be learned using SAC, we can also conclude that this phenomenon also cannot be explained through the primacy bias.
- Based on the detailed explanation above, we stress that the interpretation behind the capacity loss or primacy bias cannot explain our findings, and we think that the negative transfer is different from previously proposed problems.

### Regarding the ***critical*** typo in the caption of Figure 2

We sincerely apologize for the critical typo. In the caption of Figure 2, we used 10 random seeds for this experiment. Not 3 random seeds. We revised this part in the current version.

[1] Lyle et. al., “Understanding and Preventing Capacity Loss in Reinforcement Learning”, ICLR, 2022

[2] Nikishin et. al., “The Primacy Bias in Deep Reinforcement Learning”, ICML, 2022

---

### Meta-Review · Area_Chair_WXdA · 2023-12-09

**Metareview:**

The paper focuses on the issue of failure to learn a policy on a new task after a specific previous task, referred to as negative transfer. The authors argue that this phenomenon is distinct from the issue of capacity or plasticity and propose a simple approach to address this.

Due to several issues mentioned by the reviewers and summarized below, the current paper does not meet the bar of publication.

Strengths:

- Extensive experiments are provided on various tasks and baselines.

Weaknesses:

- For a phenomenon well-known for a while and despite giving extensive new evidence of the phenomenon on a new set of tasks, the work falls short of bringing new insights into how this issue occurs.

- In a similar manner, the work falls short of providing a reasonable solution to the issue. The proposed Reset and Distill (R&D) method is designed to sidestep negative transfer as opposed to addressing it and does not address positive transfer, which the authors also admit.

- The authors mention in the paper that “ the complexity of the offline actor
distillation is essentially on par with that of supervised learning, introducing just a minor additional computational workload.” However, the extra computations introduced by R&D and described in Algorithm 1 look anything but minor. A comparison of computational complexity in terms of wall clock time would shed light on this issue.

- As a reviewer pointed out and the authors also admit, there was a misunderstanding and mistaken application of an existing approach, Progress and Compress (P&C). Although the authors reran the experiments based on the feedback, such a significant issue requires another set of thorough reviews.

We highly encourage the reviewers to enhance the work by bringing clearer insights into the topic, which may lead to a more reasonable method without sidestepping the issue.

**Justification For Why Not Higher Score:**

In the list of weaknesses above, I clarified how the work has significant shortcomings regarding lack of insights, lack of effective solutions, and an error in experiments.

**Justification For Why Not Lower Score:**

N/A

---

### Decision · Program_Chairs · 2024-01-16

Reject